# Isoform cell-type specificity in the mouse primary motor cortex

A. Sina Booeshaghi[1], Zizhen Yao[2], Cindy van Velthoven[2], Kimberly Smith[2], Bosiljka Tasic[2], Hongkui Zeng[2] & Lior Pachter[3,4]✉

Full-length SMART-seq[1] single-cell RNA sequencing can be used to measure gene expression at isoform resolution, making possible the identification of specific isoform markers for different cell types. Used in conjunction with spatial RNA capture and gene-tagging methods, this enables the inference of spatially resolved isoform expression for different cell types. Here, in a comprehensive analysis of 6,160 mouse primary motor cortex cells assayed with SMART-seq, 280,327 cells assayed with MERFISH[2] and 94,162 cells assayed with 10x Genomics sequencing[3], we find examples of isoform specificity in cell types—including isoform shifts between cell types that are masked in gene-level analysis—as well as examples of transcriptional regulation. Additionally, we show that isoform specificity helps to refine cell types, and that a multi-platform analysis of single-cell transcriptomic data leveraging multiple measurements provides a comprehensive atlas of transcription in the mouse primary motor cortex that improves on the possibilities offered by any single technology.

Transcriptional and post-transcriptional control of individual isoforms of genes is crucial for neuronal differentiation[4–8], and isoforms of genes have been shown to be specific to cell types in mouse and human brains[9–14]. It is therefore not surprising that dysregulation of splicing has been shown to be associated with several neurodevelopmental and neuropsychiatric diseases[6,15,16]. Thus, it is of interest to study gene expression in the brain at single-cell and isoform resolution.

Nevertheless, current single-cell studies aiming to characterize cell types in the brain using single-cell RNA sequencing (scRNA-seq) have relied mostly on gene-level analysis. This is, in part, owing to the nature of the data produced by the highest-throughput single-cell methods. Popular high-throughput assays such as Drop-seq[17], 10x Genomics Chromium[3] and inDrops[18] produce 3′-end reads that are, in initial pre-processing, collated by gene to produce per-cell gene counts. SMART-seq[19] is an scRNA-seq method that produces full-length reads, enabling the quantification of individual isoforms of genes with the expectation-maximization algorithm[20]. However, such increased resolution comes at the cost of throughput. SMART-seq requires cells to be deposited in wells, thus limiting the throughput of the assay. In addition, SMART-seq requires more sequencing per cell[21].

The trade-offs are evident in analysis of scRNA-seq data from the primary motor cortex (MOp) produced by the BRAIN Initiative Cell Census Network (BICCN)[22]. We examined 6,160 (filtered) SMART-seq v4 cells and 94,162 (filtered) 10x Genomics Chromium (10xv3) cells (Extended Data Fig. 1, Fig. 2a, b) and found that while 10xv3 and SMART-seq are equivalent in defining broad classes of cell types, 3′-end technology that can assay more cells can identify some rare cell types that are missed at lower cell coverage (Extended Data Fig. 2a). Overall, 56 clusters with gene markers could be identified in the 10xv3 data but not in the SMART-seq data, whereas only 39 clusters with gene markers

could be identified in the SMART-seq data and not the 10xv3 data—this differential is consistent with previously reported comparisons of 10x Genomics Chromium and SMART-seq clusters[21,23]. However, while SMART-seq has lower throughput than some other technologies, it has a notable advantage: because it probes transcripts across their entire length, SMART-seq makes possible isoform quantification and the detection of isoform markers for cell types that cannot be detected with 3′-end technologies (Extended Data Figs. 2b, c). Moreover, the uniformity of read coverage of SMART-seq data[1] and its quantification with state-of-the-art tools[24] yields higher sensitivity than other methods, which can make possible refined cell-type classification.

To take advantage of the complementary strengths of these different platforms, we introduce an approach to scRNA-seq that links the SMART-seq resolved isoforms to the 10x Chromium defined cell types, and merges this information with spatial transcriptomic measurements obtained by MERFISH[25] (Fig. 1). In addition to revealing extensive isoform diversity and cell-type specificity in the MOp, we find evidence for previously missed transcriptionally distinct cell subtypes in the MOp. Our results extend the notion of a single-cell database beyond a list of gene markers, and we produce a gene-isoform-space single-cell atlas of the MOp using the combined 10xv3, SMART-seq and MERFISH data. Our methods are open source, reproducible, easy to use and constitute an effective workflow for leveraging full-length scRNA-seq data in combination with data from other technologies.

## Isoform markers for cell types

To identify isoform markers of cell types, we first sought to visualize our SMART-seq data using gene-derived cluster labels from the BICCN analysis (Methods). Rather than layering cluster labels on cells mapped

[1]Department of Mechanical Engineering, California Institute of Technology, Pasadena, CA, USA. [2]Allen Institute for Brain Science, Seattle, WA, USA. [3]Division of Biology and Biological Engineering, California Institute of Technology, Pasadena, CA, USA. [4]Department of Computing and Mathematical Sciences, California Institute of Technology, Pasadena, CA, USA. ✉e-mail: lpachter@caltech.edu

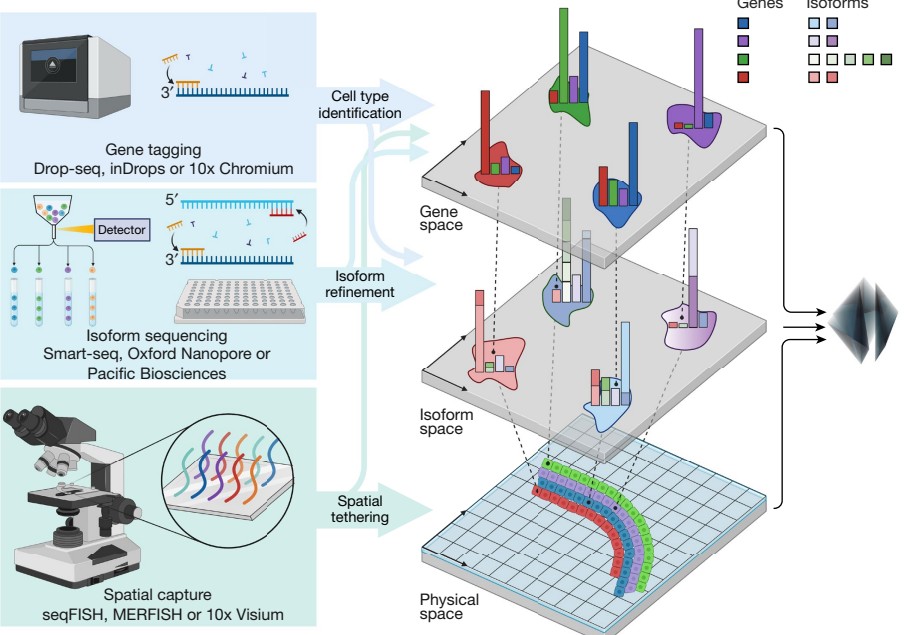

**Fig. 1 | Measuring RNA with multiple platforms.** RNA is measured using gene-tagging techniques such as the 10x Chromium scRNA-seq protocol, isoform sequencing techniques such as SMART-seq, and spatial RNA-capture techniques such as MERFISH. High-cell-throughput gene tagging enables cell-type identification with marker genes and deep full-length isoform sequencing enables cell-type marker refinement at the isoform level. Spatial RNA capture coupled with gene tagging and isoform sequencing enables spatial resolution of cell-type-specific isoform markers. The multi-method procedure for sampling RNA enables inference of spatially resolved cell-type-specific isoforms that no single technique could achieve independently[51].

to 2D with an unsupervised dimensionality-reduction technique such as *t*-distributed stochastic neighbour embedding[26] (*t*-SNE) or uniform manifold approximation and projection[27] (UMAP), we used a supervised learning approach to project cells so that they are best separated according to BICCN consortium[22] annotations using neighbourhood component analysis (NCA). This method produces meaningful representation of the global structure of the data (Fig. 2b), without overfitting (Supplementary Fig. 1a). Analysis of the projections revealed a batch effect in the 10xv3 data—which we addressed by restricting analysis to a single batch—and minimal evidence of a batch effect in the MERFISH data (Methods, Supplementary Fig. 2a, b).

Next, motivated by the discovery of genes exhibiting differential exon usage between glutamatergic and GABAergic (γ-aminobutyric acid-producing) neurons in the primary visual cortex[14], we performed a differential analysis between these two classes of neurons. We searched for significant shifts in isoform abundances in genes whose expression was stable across cell types (Methods). We discovered 398 such isoform markers belonging to 310 genes (Supplementary Table 1). Figure 2c shows an example of such an isoform from the oxidative resistance 1 (*Oxr1*) gene, which is known to be essential for protection against oxidative stress-induced neurodegeneration[28,29]. While we see no change in gene expression of *Oxr1* between these two neuron types, we find that among the 16 isoforms of the gene, one of them, *Oxr1-204*, is more highly expressed in glutamatergic neurons. The *Oxr1* gene undergoes an isoform shift in GABAergic neurons, where the expression of the *Oxr1-204* isoform is significantly lower, suggesting distinct subcellular isoform localization in the two neuron types[30]. A gene-level analysis is blind to this isoform shift (Fig. 2c, top right).

We hypothesized that there are genes exhibiting cell-type isoform specificity at all levels of the MOp cell ontology. However, detection of such genes and their associated isoforms requires meaningful cell-type assignments and accurate isoform quantifications. To assess the reliability of the SMART-seq clusters produced by the BICCN[31], we examined the correlation in gene expression by cluster with an orthogonal scRNA-seq technology, the 10xv3 3′-end assay. We clustered 94,162

10xv3 cells, also derived from the MOp, using the same method as the SMART-seq cells (Methods). The clustering method generates three hierarchies of cells: classes, subclasses and clusters. The SMART-seq data have 2 major classes (glutamatergic and GABAergic), 18 subclasses that subdivide the classes, and 62 clusters that subdivide the subclasses. The 10x data similarly contain three hierarchies of cells: two major classes (glutamatergic and GABAergic), 21 subclasses and 85 clusters. We found high correlation of gene expression between the two assays at the subclass and cluster levels (Extended Data Fig. 3).

Next, we assessed the accuracy of the SMART-seq isoform quantification and its concordance with 10xv3 quantifications of isoforms. Since not all isoforms can be quantified from 10xv3 3′-end data, we examined only isoforms containing some unique 3′ UTR sequence. This enabled us to validate the isoform quantifications using a different method (Methods). To extract isoform quantifications from 10xv3 data in cases where there was a unique 3′ sequence, we relied on transcript compatibility counts[32] produced by pseudoalignment with kallisto[24]. We were able to validate the SMART-seq isoform shift predictions at both the subclass and cluster levels (Extended Data Fig. 4). The isoform abundance correlations are slightly lower than those for gene abundance estimates (Extended Data Fig. 3), but sufficiently accurate to identify significant isoform shifts, consistent with benchmarks showing that isoforms can be quantified accurately from full-length bulk RNA-seq[33].

Having validated the cluster assignments and isoform abundance estimates, we tested for isoform switches for 16 cell subclasses excluding low quality cells (example in Fig. 2d), and then for 48 distinct clusters for subclasses that have more than one cluster (example in Fig. 2e) and more than 5 cells per cluster (Methods). At the higher level of 16 cell subclasses, we found a total of 654 isoforms from 550 genes within the glutamatergic class and 381 isoforms from 332 genes within the GABAergic class exhibiting isoform shifts among the 16 cell subclasses despite constant gene abundance (Supplementary Table 2a, b). There are several notable examples of isoform shifts at this level. For example, we find a shift in the *Snap25-202* isoform, whose expression has been specifically shown to be correlated with age and to differentially

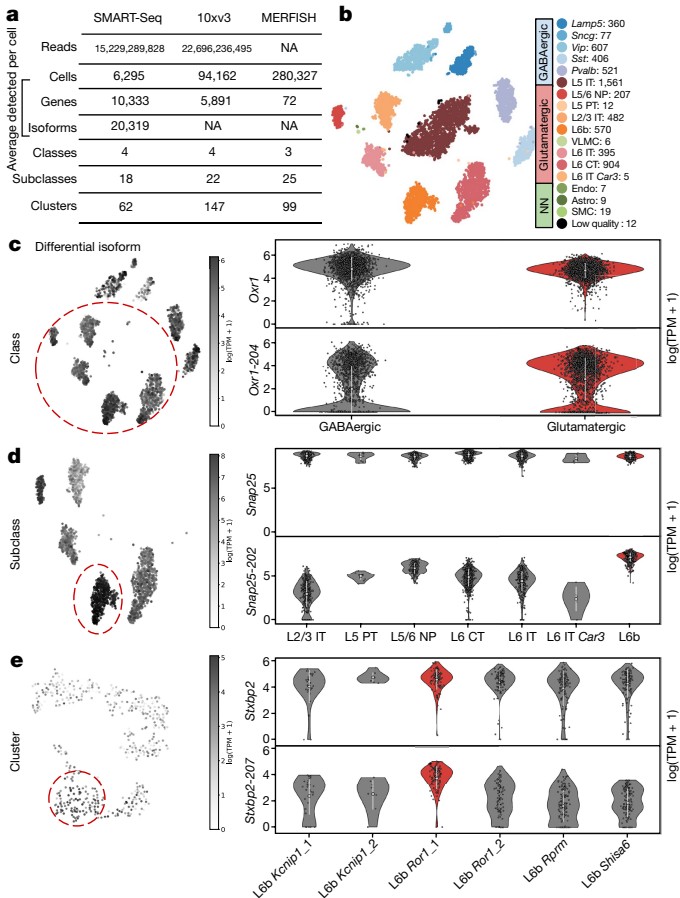

**Fig. 2 | Isoform specificity in the absence of gene specificity. a**, Overview of the data analysed. The clustering method used by the BICCN consortium generates three hierarchies of cells: classes, subclasses and clusters. NA, not applicable. **b**, A t-SNE of 10 neighbourhood components of 6,160 SMART-seq cells coloured according to subclass. Astro, astrocytes; CT, corticothalamic; endo, endothelial; IT, intratelencephalic; NN, non-neuronal; NP, near-projecting; PT, pyramidal tract; SMC, smooth muscle cells; VLMC, vascular lepotomeningeal cells. **c**, Example of a gene with an isoform specific to the glutamatergic class. The *Oxr1-204* isoform distribution in log1p(transcripts per million (TPM)) across cells (left) superimposed on the t-SNE of the cells. The cells belonging to the glutamatergic class are circled. The violin plots of the gene and isoform distributions show that the gene is not differentially expressed but the isoform is (right). **d**, Example of a gene with an isoform specific to the L6b subclass. The *Snap25-202* isoform distribution across cells (left) superimposed on the t-SNE of the cells. The cells belonging to the L6b subclass are circled. The violin plots of the gene and isoform distributions show that the gene is not differentially expressed but the isoform is (right). **e**, Example of a gene with an isoform specific to the L6b *Ror1*_1 cluster. The *Stxbp2-207* isoform distribution in log1p(TPM) across cells (left) superimposed on the t-SNE of the cells. The cells belonging to the L6b *Ror1*_1 cluster are circled. The violin plots of the gene and isoform distributions show that the gene is not differentially expressed but the isoform is (right). *$P < 0.01$ between the group and its complement. In violin plots, white circles represent the mean and white bars represent the s.d.

regulate synaptic transmission and synaptic plasticity at central synapses[34,35]. This isoform marks the L6b subclass (Fig. 2d). At the cluster level, we found 923 isoforms from 823 genes exhibiting isoform shifts among the 48 clusters passing filter despite constant gene abundance (Supplementary Table 3). Another isoform of note that marks the L6b *Ror1*_1 cluster, a subset of cells in the L6b subclass, is the *Stxbp2-207* isoform whose gene *Stxbp2* has previously been detected in the subthalamic nucleus and the posterior hypothalamus[36].

Assaying both male and female mice enabled us to examine sex-specific effects in all subclasses except for the L5 IT, which was excluded owing to batch effect (Supplementary Fig. 4). In total, these subclasses exhibited 418 sex-specific isoforms, averaging 40 isoforms per subclass (Supplementary Table 7). Unlike a recent study that reported a sex-specific cell type in the ventromedial nucleus of the hypothalamus[37], we do not find any sex-specific subclasses. However, we observed several autosomal isoforms that were differentially expressed between male and female mice. Among these, the *Shank1-203* isoform is differential in *Vip* neurons, a finding that refines previous data showing that *Shank1*, which has been shown to localize in Purkinje cells in the cortex[38], is a sex-specific gene whose expression is regulated by sex hormones[39].

We also investigated instances where clusters could be refined according to isoform expression. After reclustering each 10xv3-derived cluster using SMART-seq isoform quantifications (see Methods), we found that 12 clusters can be split by isoforms. Examining the L6 CT Grp_1 cluster, we find that the average effect size for differential isoforms that split the cluster into two sub-clusters is higher than that for genes (Extended Data Fig. 5). One isoform in particular, which splits the L6 CT Grp_1 cluster, is a protein-coding isoform of the amyloid precursor protein gene (*App*). Dysregulation of splicing for isoforms of *App* have been associated with disease pathogenesis in Alzheimer disease models[40]. Our findings show that isoform-level expression can help refine cell types in the mouse MOp beyond what is possible using gene-level expression estimates.

Along with isoforms detectable as differential between cell types without change in gene abundance, we identified isoform markers for the classes, subclasses, and clusters in the MOp ontology that are differential regardless of gene expression. We found 5,658 isoforms belonging to 3,132 genes that are specific to the glutamatergic and GABAergic classes (Fig. 3, Supplementary Table 4), 7,588 isoforms belonging to 4,171 genes within the glutamatergic class and 4,359 isoforms belonging to 2,614 genes within the GABAergic class exhibiting isoform shifts specific to subclasses (Supplementary Table 5a, b), and for the 48 clusters passing filter, 3,171 isoforms belonging to 2,461 genes exhibiting isoform shifts in clusters (Supplementary Table 6). Together, these form an isoform catalogue of the MOp (Supplementary Fig. 5a, b).

## Spatial isoform specificity

Spatial scRNA-seq methods are not currently well suited to directly probing isoforms of genes owing to the number and lengths of probes required—however, spatial analysis at the gene level can be refined to yield isoform-level results by extrapolating SMART-seq isoform quantifications (Fig. 3, Supplementary Fig. 5c).

Figure 4a, b shows an example of a gene, *Pvalb*, for which the SMART-seq quantification reveals that of the two isoforms of the gene, only one, *Pvalb-201*, is expressed. Moreover, this effect is specific to the *Pvalb* cell subclass (Fig. 3). In an examination of MERFISH spatial single-cell RNA-seq derived from 64 slices from the MOp region (Extended Data Fig. 6a), the *Pvalb* subclass, for which *Pvalb* is a marker, can be seen to be dispersed throughout the motor cortex spanning all layers (Extended Data Fig. 6b). While MERFISH probes only measure abundance of *Pvalb* at the gene level (Fig. 4c), extrapolation from the SMART-seq quantifications can be used to refine the MERFISH result to reveal the spatial expression pattern of the *Pvalb-201* isoform.

This extrapolation can be done systematically. To build a spatial isoform atlas of the MOp, we identified differentially expressed genes from the MERFISH data (Supplementary Table 8a, b) and for each of them checked whether there were SMART-seq isoform markers (from Supplementary Table 5a, b). An example of the result is shown in Fig. 3, which displays one gene for each cluster, together with the isoform label specific to that cluster and the spatial location of the specific cluster within a slice of the mouse MOp.

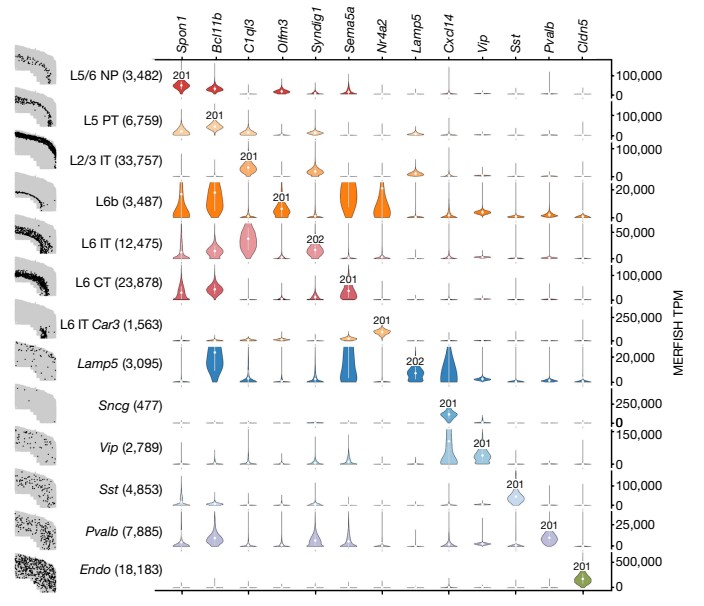

**Fig. 3 | Isoform atlas.** Spatial isoform atlas of the MOp. The scatter plots (left) show the locations of cells (black dots) in the indicated subclass within a single representative slice of the mouse MOp as assayed by MERFISH. Right, each column corresponds to a marker gene in the MERFISH dataset and each row corresponds to a subclass (number of cells in parentheses) in which one isoform (labelled on the diagonal) was differential in the SMART-seq dataset for that subclass. This spatial isoform inference links isoform expression from the SMART-seq data with the physical locations of cells expressing that isoform from the MERFISH data. The normalized gene expression values are plotted for each subclass–gene pair in TPM. The white circles in the violin plots represent the mean and the white bars represent the s.d.

We hypothesized that the mouse MOp exhibits changes in isoform expression associated with the physical location of cells[41,42]. To determine whether there are isoforms that increase or decrease in expression along the depth of the motor cortex, we first estimated the position of the various layers in the glutamatergic subclasses (Extended Data Fig. 7a, Methods), performed weighted least-squares regression on the centroids of the subclasses and inferred isoform expression from the SMART-seq data. While we find many isoforms that exhibit a significant change in expression across the depth (Extended Data Fig. 7b, c, Supplementary Table 8c), none of the isoforms that pass our filter exhibit a monotonic change with respect to the mean. This suggests that non-linear models may be better suited to study isoform variability across the depth of the mouse MOp.

While direct measurement of isoform abundance may be possible with spatial RNA-seq technologies such as SEQFISH[37] or MERFISH[2], such resolution would require dozens of probes to be assayed per gene (Supplementary Fig. 6), each of which is typically tens of base pairs in length. Thus, while isoforms can theoretically be detected in cases where they contain large stretches of unique sequence, the technology is currently prohibitive for assaying most isoforms, making the extrapolation procedure described here of practical relevance (Supplementary Table 9).

## Splicing markers

Isoform quantification of RNA-seq can be used to distinguish shifts in expression between transcripts that share transcriptional start sites and shifts due to the use of distinct transcription start sites (TSSs). Investigating such differences can, in principle, shed light on transcriptional versus post-transcriptional regulation of detected isoform shifts[43,44]. For example, in the glutamatergic class, *Ptk2b* (Extended

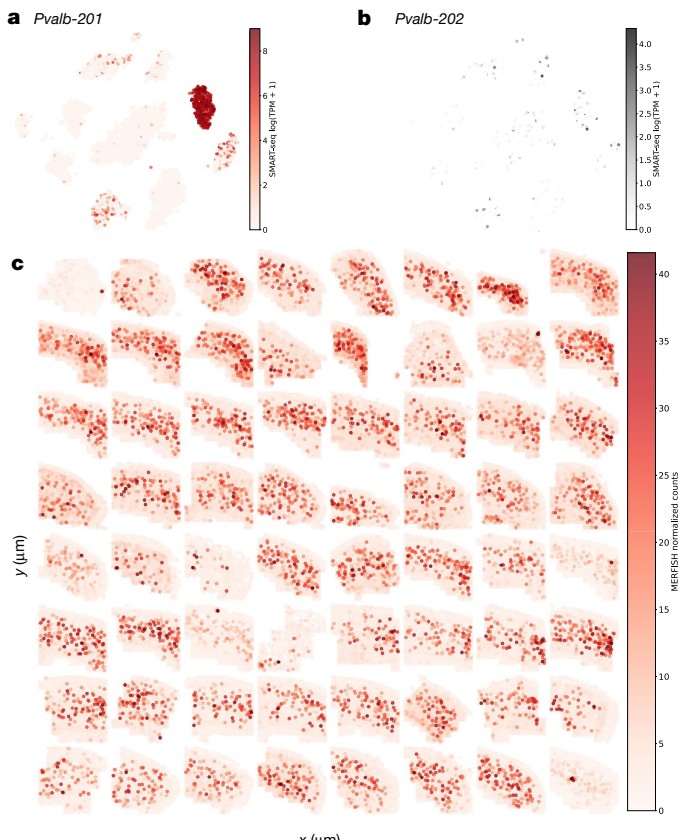

**Fig. 4 | Spatial extrapolation of isoform expression. a, b,** Expression of the *Pvalb-201* (**a**) and *Pvalb-202* (**b**) isoforms in log1p(TPM) units for each cell superimposed on the NCA–*t*-SNE plot, as assayed by SMART-seq. **c,** Spatial expression of the *Pvalb-201* isoform across 64 slices from the MOp, as extrapolated from probes for the *Pvalb* gene assayed by MERFISH. Each cell represented by MERFISH data is coloured by its expression of *Pvalb* in normalized counts.

Data Fig. 8a) exhibits differential expression of transcripts between start sites (Extended Data Fig. 8b). This gene is known to be associated with Alzheimer's disease and its transcript usage is mediated by genetic variation[45]. We find that isoforms sharing the preferential start site exhibit no discernible difference in expression (Extended Data Fig. 8c), suggesting that the observed differences result from cell-type-specific transcriptional, rather than post-transcriptional regulation. We identified 1,971 isoforms from 128 groups of TSSs where the TSSs are preferentially expressed in either GABAergic, glutamatergic or non-neuronal classes, even when the expression of isoforms contained within the TSS is constant (Supplementary Table 10a, c, d). Such cases are likely to be instances where the TSS shifts between cell types are a result of differential promoter usage—that is, the result of a transcriptional program.

We also examined post-transcriptional programs (Supplementary Table 10b), instances where the TSSs are not differential between classes, but where there are isoform shifts within TSSs between classes. We find 31 isoforms from 28 TSS groups that are differential between classes when the TSS group is not. One such example is expression of isoforms *Rtn1-201* and *Rtn1-203*, which share the same TSS in the *Rtn1* gene. The glutamatergic class exhibits preferential expression of *Rtn1-201*, which was previously shown to be expressed in grey matter[46], whereas the GABAergic class does not (Extended Data Fig. 9). These cases are likely to be instances where isoform shifts between cell types are a result of differential splicing—that is, the result of a post-transcriptional program.

## Discussion

Our spatially resolved isoform atlas of the mouse MOp expands on previously identified gene markers, extending the catalogue to isoform markers for cell types characterized by the BICCN[22]. Our approach leverages distinct strengths of different technologies, using the isoform resolution of SMART-seq in conjunction with the complementary cell depth obtainable with 10x Genomics technology and the spatial resolution produced with MERFISH to spatially place cell-type isoform markers. This validated approach, in which we leverage technologies that are broadly consistent (Extended Data Fig. 10) yet complementary in their strengths, is important because isoform specificity could help in explaining the molecular basis of morphological differences. For example, *Pvalb* cells observed in hippocampal *Pvalb* interneurons cannot be distinguished morphologically solely on the basis of gene-level analysis[9,47,48].

Spatial isoform markers also enable more targeted assays for 'automatic expression histology' and can facilitate investigation of the functional significance of cell-type isoform specificity. Recently developed experimental methods for this purpose—for example, isoform screens[49]—are a promising direction and will be key to understanding the significance of the vast isoform diversity in the brain[50].

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

## Methods

All of the results and figures in the paper are reproducible starting with the raw reads using scripts and code downloadable from https://github.com/pachterlab/BYVSTZP_2020. The repository makes the method choices completely transparent, including all parameters and thresholds used. All P-values were corrected using Bonferroni correction and all error bars denote ±1× s.d. from the mean.

### Tissue collection and isolation of cells

**Mouse breeding and husbandry.** All procedures were carried out in accordance with Institutional Animal Care and Use Committee protocols at the Allen Institute for Brain Science. Mice were provided food and water ad libitum and were maintained on a regular 12-h day:night cycle at no more than five adult animals per cage. For this study, we enriched for neurons by using Snap25-IRES2-Cre mice[52] (MGI:J:220523) crossed to Ai14[53] (MGI: J:220523), which were maintained on the C57BL/6J background (RRID:IMSR_JAX:000664). Mice were euthanized at 53–59 days of postnatal age. Tissue was collected from both males and females for scRNA SMART and scRNA 10x v3 analysis.

**Single-cell isolation.** We isolated single cells by adapting previously described procedures[14,31]. The brain was dissected, submerged in artificial cerebrospinal fluid (ACSF)[31], embedded in 2% agarose and sliced into 250-μm (SMART-seq) or 350-μm (10x Genomics) coronal sections on a compresstome (Precisionary Instruments). The Allen Mouse Brain Common Coordinate Framework version 3 (CCFv3, RRID:SCR_002978)[54] ontology was used to define MOp for dissections.

For SMART-seq, MOp was microdissected from the slices and dissociated into single cells with 1 mg ml$^{-1}$ pronase (Sigma P6911-1G) and processed as previously described[31]. For 10x Genomics analysis, tissue pieces were digested with 30 U ml$^{-1}$ papain (Worthington PAP2) in ACSF for 30 min at 30 °C. Enzymatic digestion was quenched by exchanging the papain solution three times with quenching buffer (ACSF with 1% FBS and 0.2% BSA). The tissue pieces in the quenching buffer were triturated through a fire-polished pipette with 600-μm diameter opening approximately 20 times. The solution was allowed to settle and supernatant containing single cells was transferred to a new tube. Fresh quenching buffer was added to the settled tissue pieces, and trituration and supernatant transfer were repeated using 300-μm and 150-μm fire polished pipettes. The single-cell suspension was passed through a 70-μm filter into a 15-ml conical tube with 500 μl high-BSA buffer (ACSF with 1% FBS and 1% BSA) at the bottom to help cushion the cells during centrifugation at 100g in a swinging-bucket centrifuge for 10 min. The supernatant was discarded and the cell pellet was resuspended in quenching buffer.

All cells were collected by fluorescence-activated cell sorting (BD Aria II, RRID: SCR_018091) using a 130-μm nozzle. Cells were prepared for sorting by passing the suspension through a 70-μm filter and adding DAPI (to the final concentration of 2 ng ml$^{-1}$). The sorting strategy was as previously described[31], with most cells collected using the tdTomato-positive label. For SMART-seq, single cells were sorted into individual wells of 8-well PCR strips containing lysis buffer from the SMART-seq v4 Ultra Low Input RNA Kit for Sequencing (Takara 634894) with RNase inhibitor (0.17 U μl$^{-1}$), immediately frozen on dry ice, and stored at −80 °C. For 10x Genomics, 30,000 cells were sorted within 10 min into a tube containing 500 μl of quenching buffer. Each aliquot of 30,000 sorted cells was gently layered on top of 200 μl of high-BSA buffer and immediately centrifuged at 230g for 10 min in a swinging-bucket centrifuge. Supernatant was removed and 35 μl of buffer was left behind, in which the cell pellet was resuspended. The cell concentration was quantified, and immediately loaded onto the 10x Genomics Chromium controller.

### Genomic library preparation and sequencing

For SMART-seq library preparation, we performed the procedures with positive and negative controls as previously described[31]. The SMART-seq v4 (SSv4) Ultra Low Input RNA Kit for Sequencing (Takara 634894) was used to reverse transcribe poly(A) RNA and amplify full-length cDNA. Samples were amplified with 18 cycles in 8-well strips, in sets of 12–24 strips at a time. All samples proceeded through Nextera XT DNA Library Preparation (Illumina Cat# FC-131-1096) using Nextera XT Index Kit V2 (Illumina FC-131-2001) and a custom index set (Integrated DNA Technologies). Nextera XT DNA Library preparation was performed according to the manufacturer's instructions, with a modification to reduce the volumes of all reagents and cDNA input to 0.4× or 0.5× of the original protocol.

For 10x v3 library preparation, we used the Chromium Single Cell 3′ Reagent Kit v3 (10x Genomics 1000075). We followed the manufacturer's instructions for cell capture, barcoding, reverse transcription, cDNA amplification, and library construction. We targeted sequencing depth of 120,000 reads per cell.

Sequencing of SMART-seq v4 libraries was performed as described[31]. In brief, libraries were sequenced on an Illumina HiSeq2500 platform (paired-end with read lengths of 50 bp). The 10x v3 libraries were sequenced on Illumina NovaSeq 6000 (RRID:SCR_016387).

### Pre-processing single-cell RNA-seq data

The 6,295 SMART-seq cells were processed using kallisto with the 'kallisto pseudo' command[24]. The 94,162 10x Genomics v3 cells were pre-processed with kallisto and bustools[55]. Gene-count matrices were made by using the -genecounts flag and TCC matrices were made by omitting it. The mouse transcriptome reference used was GRCm38.p3 (mm10) RefSeq annotation gff file retrieved from NCBI on 18 January 2016 (https://www.ncbi.nlm.nih.gov/genome/annotation_euk/all/), for consistency with the reference used by the BICCN consortium[22].

The GTF and the GRCm38 genome fasta file (https://github.com/pachterlab/BYVSTZP_2020/releases/tag/biorxiv_v1), provided by the BICCN consortium, were used to create a transcriptome fasta file, transcripts-to-genes map, and kallisto index using kb ref -i index.idx, -g t2g.txt -f1 transcriptome.fa genome.fa genes.gtf. To validate the SMART-seq isoform quantifications, we first examined the robustness of the quantifications to gene annotation, and found an average correlation at the isoform level of 0.965 between the BICCN-derived quantifications we used in our analysis[22] and the mouse GENCODE M25-derived quantifications (Supplementary Fig. 3). The GENCODE M25 mouse transcriptome reference (https://github.com/pachterlab/BYVSTZP_2020/blob/master/reference/gencode/fasta_link.txt) and the kallisto index were built using kallisto index -i index.idx gencode.vM25.transcripts.fa.gz.

Isoform and gene-count matrices were generated for the Smart-seq2 data using the kallisto pseudo command. Cluster assignments were associated with cells using cluster labels generated by the BICCN consortium[22]. The labels are organized in a hierarchy of three levels: classes, subclasses and clusters. The cluster labels for the cells can be downloaded from https://github.com/pachterlab/BYVSTZP_2020.

### Clustering and cell type assignment

Our analyses used SMART-seq cell labels produced by the BICCN[22]. In brief, the assignment of cell types to the SMART-seq cells was based on an extension of the cluster merging algorithm in the scratch.hicat package[31]. The clustering method generates three hierarchies of cells: classes, subsets of cells within classes called subclasses, and subsets of cells within subclasses called clusters.

To build a common adjacency matrix incorporating samples from all the datasets, first a subset of datasets (reference datasets) was selected. The 10x v2 single cell dataset from Allen (scRNA 10x v2 A) and 10x v3 single nucleus dataset from Broad (snRNA 10x v3 B) were used as references.

The key steps of the pipeline are as follows: (1) perform single-dataset clustering, (2) select anchor cells for each reference dataset, (3) select highly variable genes (HVG), (4) compute k-nearest neighbors,

(5) compute the Jaccard similarity, (6) perform Louvain clustering, (7) merge clusters, (8) cluster iteratively, and (9) compile and merge clusters. Further details are in ref. [22].

### Normalization and filtering of SMART-seq data

Isoform counts were first divided by the length of transcript to obtain abundance estimates proportional to molecule copy numbers. Since reads can come from anywhere in the transcriptome, it is likely that longer isoforms are enriched. Therefore normalizing isoform abundances by length is crucial to accurately estimating mRNA copy number. This has been shown in numerous studies on the accurate estimate of isoform abundance[56,57].

After normalizing by length, we then removed isoforms that had fewer than one count and that were in fewer than one cell. We also removed genes and their corresponding isoforms that had a dispersion of less than 0.001.

To generate the cell-by-gene matrix we summed the isoforms that correspond to the same gene. Cells with fewer than 250 gene counts and with greater than 10% mitochondrial content were removed. Cells were normalized to TPM by dividing the counts in each cell by the sum of the counts for that cell, then multiplying by 1,000,000. The count matrices were then transformed with log1p and the columns were scaled to unit variance and zero mean. The resulting gene and isoform matrix contained 6,160 cells and 19,190 genes, corresponding to 69,172 isoforms.

Highly variable isoforms and genes were identified by first computing the dispersion for each feature, and then binning all of the features into 20 bins. The dispersion for each feature was normalized by subtracting the mean dispersion and dividing by the variance of the dispersions within each bin. Then the top 5,000 features were retained based on the normalized dispersion. This was computed[58] using scanpy.pp.highly_variable_genes with n_top_genes = 5000, flavor=seurat, and n_bins=20.

### Normalization and filtering of 10xv3 data

To generate the cell-by-gene matrix we used 'bustools count –genecounts'. The cell-by-isoform matrix was generated using 'bustools count' and restricting to the equivalence classes that contained only one isoform thus generating a cell-by-isoform matrix. Both matrices were loaded into python using kb python. Cells with less than 250 gene counts and with greater than 21.5% mitochondrial content were removed. Cells were normalized to counts per million (CPM) by dividing the counts in each cell by the sum of the counts for that cell, then multiplying by 1,000,000. The count matrices were then transformed with log1p and the columns scaled to unit variance and zero mean. The resulting gene matrix contained 94,162 cells and 24,575 genes. We removed the cells that were identified as low quality by the BICCN consortium. We identified batch effect among cells assayed on different dates so we restricted our analysis to only the cells assayed on the same date and selected the date with the most number of cells (Supplementary Fig. 2). Additionally, we performed pairwise comparison of gene counts for each of the 4 10xv3 batches and found the Pearson correlation to be very high for all pairs, with a mean of 0.9979 indicating limited batch effect between batches assayed on the same date.

Highly variable isoforms and genes were identified by first computing the dispersion for each feature, and then binning all of the features into 20 bins. The dispersion for each feature was normalized by subtracting the mean dispersion and dividing by the variance of the dispersions within each bin. Then the top 5,000 features were retained based on the normalized dispersion. This was computed[58] scanpy.pp.highly_variable_genes with n_top_genes = 5000, flavor=seurat, and n_bins=20.

### Dimensionality reduction and visualization

To visualize the SMART-seq data with predefined cluster labels produced via a joint analysis with many other data types we performed NCA[59] on the full scaled log(TPM + 1) matrix using the subcluster labels, to ten components. t-SNE[26] was then performed on the 10 NCA components. NCA takes as input not just a collection of cells with their associated abundances, but also cluster labels for those cells, and seeks to find a projection that minimizes leave-one-out k-nearest neighbour error[59]. Moreover, t-SNE applied to PCA (Supplementary Fig. 1b) scrambles the proximity of glutamatergic and GABAergic cell types, while t-SNE of NCA appears to respect global structure of the cells. While UMAP applied to PCA of the data (Supplementary Fig. 1c) appears to be better than t-SNE in terms of preserving global structure, it still does not separate out the cell types as well as NCA (Supplementary Fig. 1d). t-SNE was computed using sklearn.manifold. t-SNE was generated with default parameters and random state 42. Similarly uniform manifold approximation was performed on the 10 NCA components and the 50 truncated singular value decomposition (SVD)-derived components. UMAP[27] was computed with the umap package with default parameters.

To ensure that NCA was not overfitting cells to their corresponding subclasses, we randomly permuted all of the subclasses labels and reran the NCA-to-t-SNE dimensionality-reduction method. We observed uniform mixing of the permuted subclass labels, indicating that NCA was not overfitting the cells to their corresponding subclasses.

### Sample size

No explicit calculations were performed to determine sample size. We analysed 6,160 mouse MOp cells assayed with SMART-seq, 280,327 cells assayed with MERFISH, and 94,162 cells assayed with 10x Genomics Chromium v3. We analysed both male and female mice to understand differences in gene and isoform expression. The sample size for differential expression was set to be such that 90% of cells in a cluster have a non-zero expression of the tested gene. The smallest cluster size contained seven cells, with all cells having non-zero expression of the tested genes. We computed error bars for all tests to ensure that sample sizes were sufficient.

### Batch effects

After finding a meaningful projection that appears to respect global structure of the cells we searched for possible sources of batch effect within the datasets. We found evidence of batch effect in the 10xv3 data by assay date (Supplementary Fig. 2a). To ensure that our findings were not confounded by this batch effect we selected the set of cells from only one assay date and picked the set with the largest number of cells and the one with cells present in all clusters. We then looked at the MERFISH data and found minimal evidence of batch effect across samples based on the distribution of batch labels across clusters where the observed fraction of cells per batch in each cluster was almost exactly the expected fraction of cells per batch assuming uniform mixing (Supplementary Fig. 2b).

In further examining the single 10xv3 batch we settled on, we noted a low correlation in one case, the L5 IT subclass. The low correlation was also observed in a comparison between SMART-seq and MERFISH gene expression data (Extended Data Fig. 10a), and 10xv3 and MERFISH data (Extended Data Fig. 10b). We hypothesized that this low correlation stems from a subclass-specific sex effect within the L5 IT, where those cells differ drastically in their overall expression compared to other subclasses. The L5 IT subclass contains seven clusters in the SMART-seq data, four clusters in the 10xv3 data, and four clusters in the MERFISH data.

To determine the source of the low correlation within the L5 IT between SMART-seq, 10x and MERFISH data, we examined differential genes between male and female cells within each subclass. We found that cells within the L5 IT of the 10x and SMART-seq data exhibited sex-specific segregation (Supplementary Fig. 4a, b). After performing differential expression between male and female cells within all subclasses we found that the L5 IT had the highest amount of uniquely differential genes (Supplementary Fig. 4c) and that the SMART-seq and 10x data had 37 common genes that were differentially expressed (Supplementary Fig. 4d). The other subclasses, however, did not exhibit

sex-based segregation. Without being able to rule out that the low correlation for L5 IT cells across the technologies was due to confounding between batch and sex in the dataset, we decided to excluded the subclass from our analyses.

## Measuring number of isoforms per gene

We parsed the transcripts-to-genes map, grouping together transcripts that had the same end site that were in the same gene. We then counted the number of these end site sets within a gene and plotted them against the number of isoforms within that gene.

## Cross-technology cluster correlation

The correlation between 10xv3-SMART-seq, 10xv3-MERFISH, and SMART-seq-MERFISH, was performed at the gene level and between cells grouped by subclasses for all three pairs of technologies, and at the isoform level and between cells grouped by cluster for only the 10xv3 and SMART-seq. For each pair we started with two raw matrices and restricted to the set of genes or isoforms common to the two. Then we normalized the counts for each matrix per cell to one million, log1p-transformed the entire matrix, and scaled the features to zero mean and unit variance. Within each cluster, we restricted the features to those present in at least 50% of the cells. We then found the mean cell within the respective clusters in the two matrices and computed the Pearson correlation between them. These methods were implemented for Extended Data Figs. 3, 4, 10. In terms of accuracy of different technologies, we found good agreement between quantifications from SMART-seq, 10xv3 and MERFISH (Extended Data Fig. 10).

Comparisons of different scRNA-seq technologies have tended to focus on throughput, cost and gene-level accuracy[60] in a winner-takes-all competition. Our results shed some light on the matter; it has been previously shown that quantification of isoform abundance is necessary for accurate gene-level estimates[61], and we found that it matters in practice (Supplementary Fig. 7, Supplementary Tables 11a, b, 12a, b). This highlights the importance of proper isoform quantification of SMART-seq data, even for gene-centric analysis[56,60–63], when used in conjunction with 10x Genomics and MERFISH data.

## Isoform atlas

For each level of clustering, class, subclass and cluster, we performed a t-test for each gene or isoform between the cluster and its complement on the log1p counts. To identify isoform enrichment that is masked at a gene-level analysis, we looked for isoforms that were upregulated by checking that the gene containing that isoform was not significantly expressed in that cluster relative to the complement of that cluster. Isoforms that were expressed in less than 90% of the cells in that cluster were ignored. All t-tests used a significance level of 0.01 and all P-values were corrected for multiple testing using Bonferroni correction.

## MERFISH isoform extrapolation

First, we identified the genes that mark the specific subclass within the MERFISH data. The *Pvalb* gene is a marker for the *Pvalb* subclass. Then we performed differential analysis on the SMART-seq data at the isoform level on the subclasses to identify the isoforms that mark each of the SMART-seq subclasses. Only one of the two isoforms for *Pvalb* marked the *Pvalb* cluster. This allowed us to extrapolate the fact that the specific *Pvalb* isoform is being detected in the MERFISH data.

Additionally, we identified all of the genes that mark the specific subclasses in the MERFISH data through differential analysis and checked if their underlying isoforms were also differentially expressed. We then noted which isoforms were differentially expressed for the spatial isoform atlas.

## Weighted least-squares regression

First, we selected a representative slice of the MOp. Then we found the outer hull of the MOp by using scipy.spatial.ConvexHull. We selected the points that defined the upper boundary of the Mop, then performed linear regression to fit a line to those points using sklearn.linear_model. LinearRegression(). For each subclass in the glutamatergic class of cells we identified the centroid of the subclass and determined the perpendicular distance of the centroid to the MOp boundary line. We normalized the set of distances by dividing by the centroid with the largest distance to the boundary.

We look at the isoforms for which all of the subclasses had non-zero expression in at least 90% of cells. For each isoform, we performed weighted least squares regression for all of the subclasses with the weights equal to the variance of isoform expression for each subclass. We used the statsmodel.api.sm.WLS function. All weighted least-squares tests used a significance level of 0.01 and all F-score P-values were corrected for multiple testing using Bonferroni correction. Monotonicity was checked for isoforms with an absolute value slope greater than 1.5.

## Grouping transcripts by start site

Using the transcripts to genes map and the filtered isoform matrix, we grouped isoforms by their TSS into TSS classes and summed the raw counts for the isoforms within each TSS class to create a cell-by-TSS matrix. Differential analysis was then performed in exactly the same way as above. For each cluster and each TSS or isoform, a t-test was performed between the cells in that cluster and the cells in the complement of that cluster. All statistical tests used a significance level of 0.01 and all P-values were corrected for multiple testing using Bonferroni correction.

## Comparison of naive and EM quantification

Naive gene-count matrices were constructed from the SMART-seq data by summing the counts corresponding to a single gene. Gene-count matrices quantified by the expectation maximization (EM) algorithm and normalized appropriately were made with SMART-seq by first dividing isoform abundances by the length of their transcripts, and then summing the abundances of isoforms by gene. Differential analysis was performed independently on these two gene-count matrices and the resultant differential genes were compared. Differential expression was then performed on all of the genes for both the EM and naive gene quantifications. All statistical tests used a significance level of 0.01 and all P-values were corrected for multiple testing using Bonferroni correction.

## Software versions

Software versions used were: Anndata 0.7.1, bustools 0.39.4, awk (GNU awk) 4.1.4, grep (GNU grep) 3.1, kallisto 0.46.1, kb_python 0.24.4, Matplotlib 3.0.3, Numpy 1.18.1, Pandas 0.25.3, Scanpy 1.4.5.post3, Scipy 1.4.1, sed (GNU sed) 4.4, sklearn 0.22.1, statsmodels 0.12.1, tar (GNU tar) 1.29, umap 0.3.10.

## Reporting summary

Further information on research design is available in the Nature Research Reporting Summary linked to this paper.

## Data availability

The single-cell RNA-seq data used in this study were generated as part of the BICCN consortium[22]. The 10xv3 and SMART-seq data can be downloaded from http://data.nemoarchive.org/biccn/lab/zeng/transcriptome/scell/. The MERFISH data are available at https://caltech.box.com/shared/static/dzqt6ryytmjbgyai356s1z0phtnsbaol.gz. All cell annotations and cluster labels are available at https://github.com/pachterlab/BYVSTZP_2020/tree/master/reference.

## Code availability

The software used to generate the results and figures of the paper is available at https://github.com/pachterlab/BYVSTZP_2020.

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

**Acknowledgements** We thank members of the BICCN consortium, especially the Mini-MOp analysis group, for helpful conversations related to transcriptome analysis of the MOp. We thank N. Volovich, V. Ntranos and P. Melsted for help with a preliminary quantification of the SMART-seq data. Figure 1 was created from scratch using the tools available on Biorender.com. Extended Data Fig. 6a was obtained from http://atlas.brain-map.org/atlas. This work was funded by the NIH Brain Initiative via grant U19MH114930 to H.Z. and L.P.

**Author contributions** A.S.B. and L.P. conceived the study. A.S.B. implemented the methods and produced the results and figures. A.S.B. and L.P. analysed the data and wrote the manuscript. Z.Y., C.v.V., K.S., B.T. and H.Z. produced the SMART-seq and 10xv3 data.

**Competing interests** The authors declare no competing interests.

**Additional information**
**Correspondence and requests for materials** should be addressed to Lior Pachter.

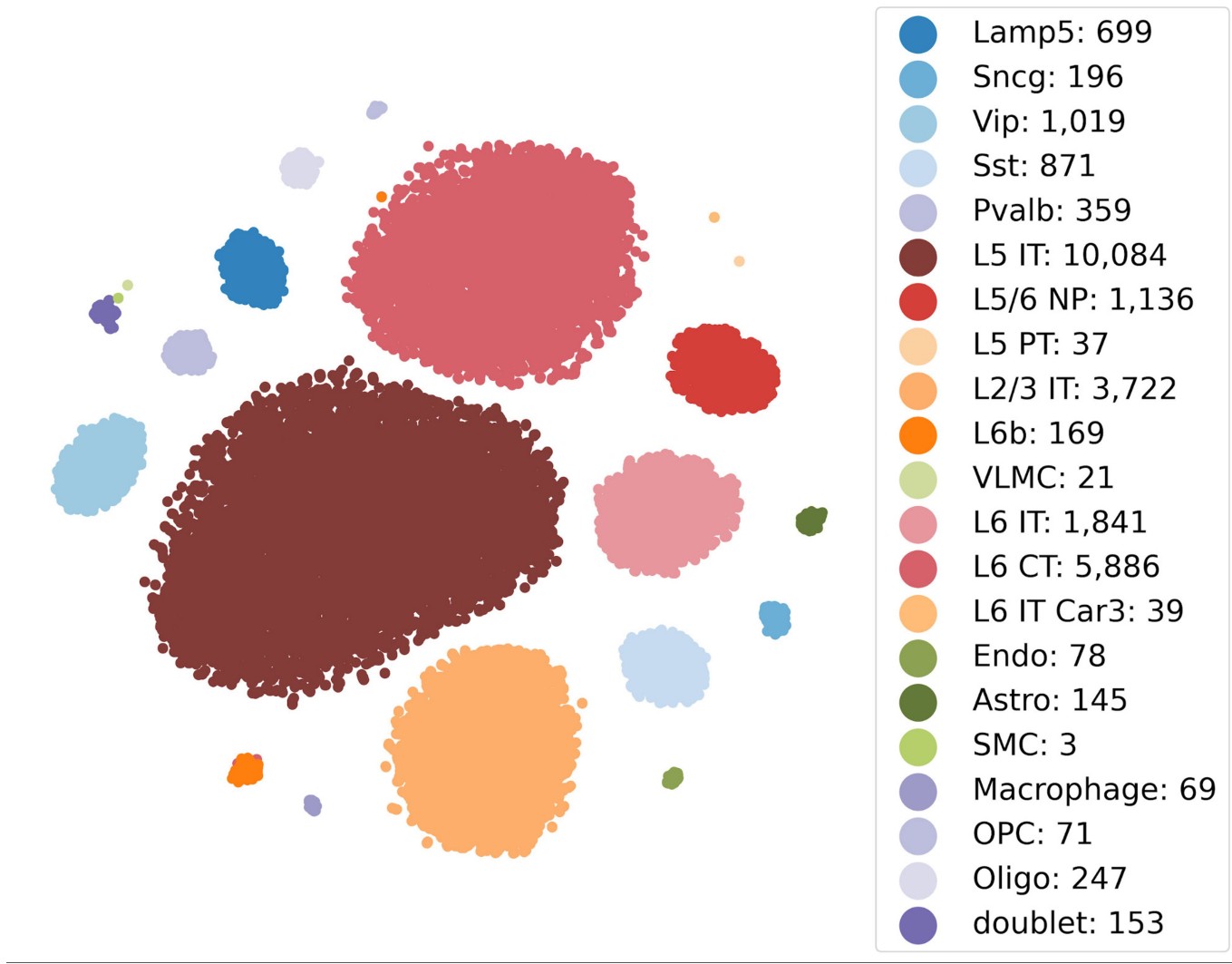

Lamp5: 699
Sncg: 196
Vip: 1,019
Sst: 871
Pvalb: 359
L5 IT: 10,084
L5/6 NP: 1,136
L5 PT: 37
L2/3 IT: 3,722
L6b: 169
VLMC: 21
L6 IT: 1,841
L6 CT: 5,886
L6 IT Car3: 39
Endo: 78
Astro: 145
SMC: 3
Macrophage: 69
OPC: 71
Oligo: 247
doublet: 153

**Extended Data Fig. 1 | 10xv3 neighborhood component analysis.**
Neighborhood component analysis (NCA) to 10 dimensions followed by
t-distributed stochastic neighbor embedding (t-SNE) of 26,845 10xv3 cells
from the mouse primary motor cortex annotated with cell subclass
assignments. The number of cells in each subclass is displayed next to the
subclass label.

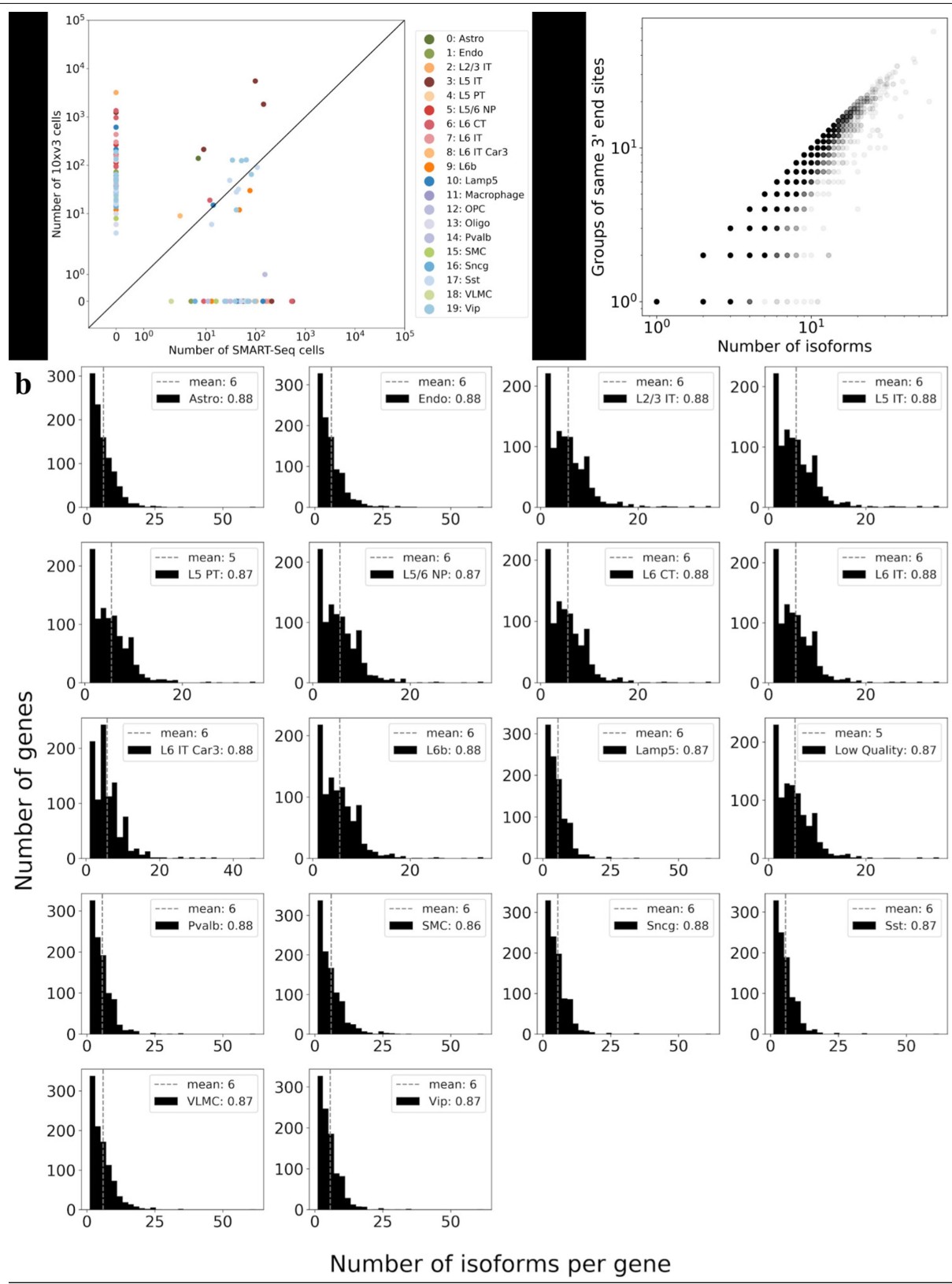

**Extended Data Fig. 2 | Cell type identification and isoform summary statistics. a**) Comparison of subclass identification for 10xv3 and SMART-seq. Each technology identified subclasses separately, but with the same method. 56 clusters with gene markers were identified in the 10xv3 data but not in the SMART-seq data while 39 clusters with gene markers were identified in the SMART-seq data and not the Chromium data. **b**) The distribution of the number of isoforms per gene within each of 18 subclasses, computed from the top 998

most highly expressed genes in the SMART-seq dataset. The number associated with each class indicates the fraction of genes for which there are more than one isoform. **c**) Extent of isoform diversity in groups of transcripts sharing a 3′ end. Each point displays the density of the number of groups (y-axis) containing a given number of isoforms (x-axis) for a single gene. Points along the line $y = x$ correspond to genes where all transcripts contained within the gene have unique 3′ ends.

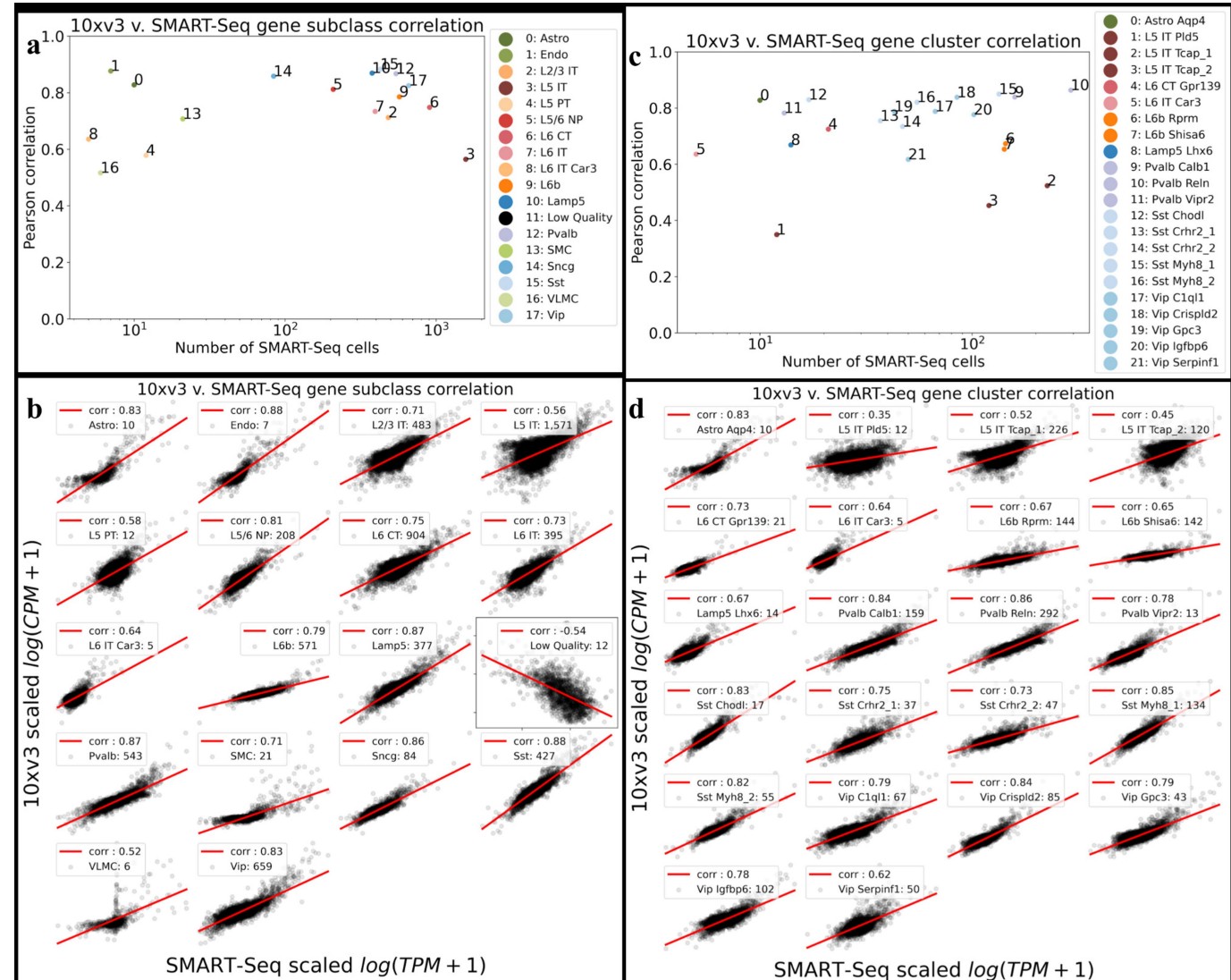

**Extended Data Fig. 3 | Gene level subclass validation with 10xv3 and SMART-seq. a)** Pearson correlation by subclass of the mean gene expression in 10xv3 and the mean gene expression in SMART-seq, against the size of the subclass, for genes that are expressed in at least 50% of cells in that subclass. **b)** Scatter plot by subclass of the mean gene expression in 10xv3 vs the mean gene expression in SMART-seq for genes that are expressed in at least 50% of cells in that subclass. The subclass sizes and Pearson correlation values are also reported. **c)** Pearson correlation for common clusters in the 10xv3 and SMART-seq datasets, computed for each cluster with respect to genes expressed in at least 50% of cells of the cluster. **d)** Scatter plot by cluster of the mean gene expression in 10xv3 vs the mean gene expression in SMART-seq, for genes that are expressed in at least 50% of cells. The cluster sizes and Pearson correlation values are also reported.

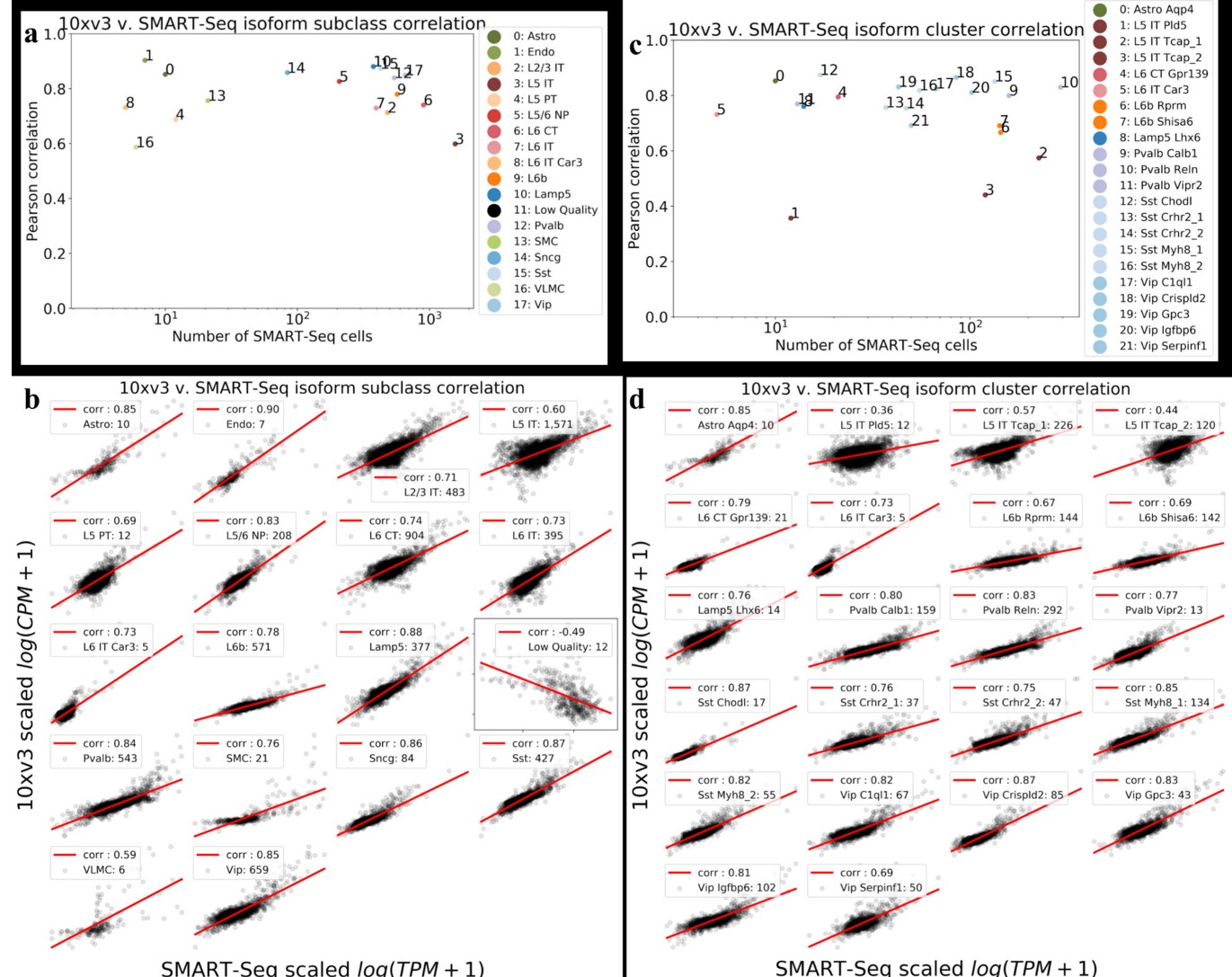

**Extended Data Fig. 4 | Isoform level subclass validation with 10xv3 and SMART-seq. a)** Pearson correlation by subclass of the mean isoform expression in 10xv3 and the mean isoform expression in SMART-seq, against the size of the subclass, for isoforms that are expressed in at least 50% of cells in that subclass. **b)** Scatter plot by subclass of the mean isoform expression in 10xv3 vs the mean isoform expression in SMART-seq, for isoforms that are expressed in at least 50% of cells in that subclass. The subclass sizes and Pearson correlation values are also reported. **c)** Pearson correlation by cluster

of the mean isoform expression in 10xv3 and the mean isoform expression in SMART-seq, against the size of the cluster, for isoforms that are expressed in at least 50% of cells in that cluster for clusters that are common to both the SMART-Seq and 10x datasets. **d)** Scatter plot by cluster of the mean isoform expression in 10xv3 vs the mean isoform expression in SMART-seq for isoforms that are expressed in at least 50% of cells in that cluster. The cluster sizes and Pearson correlation values are also reported.

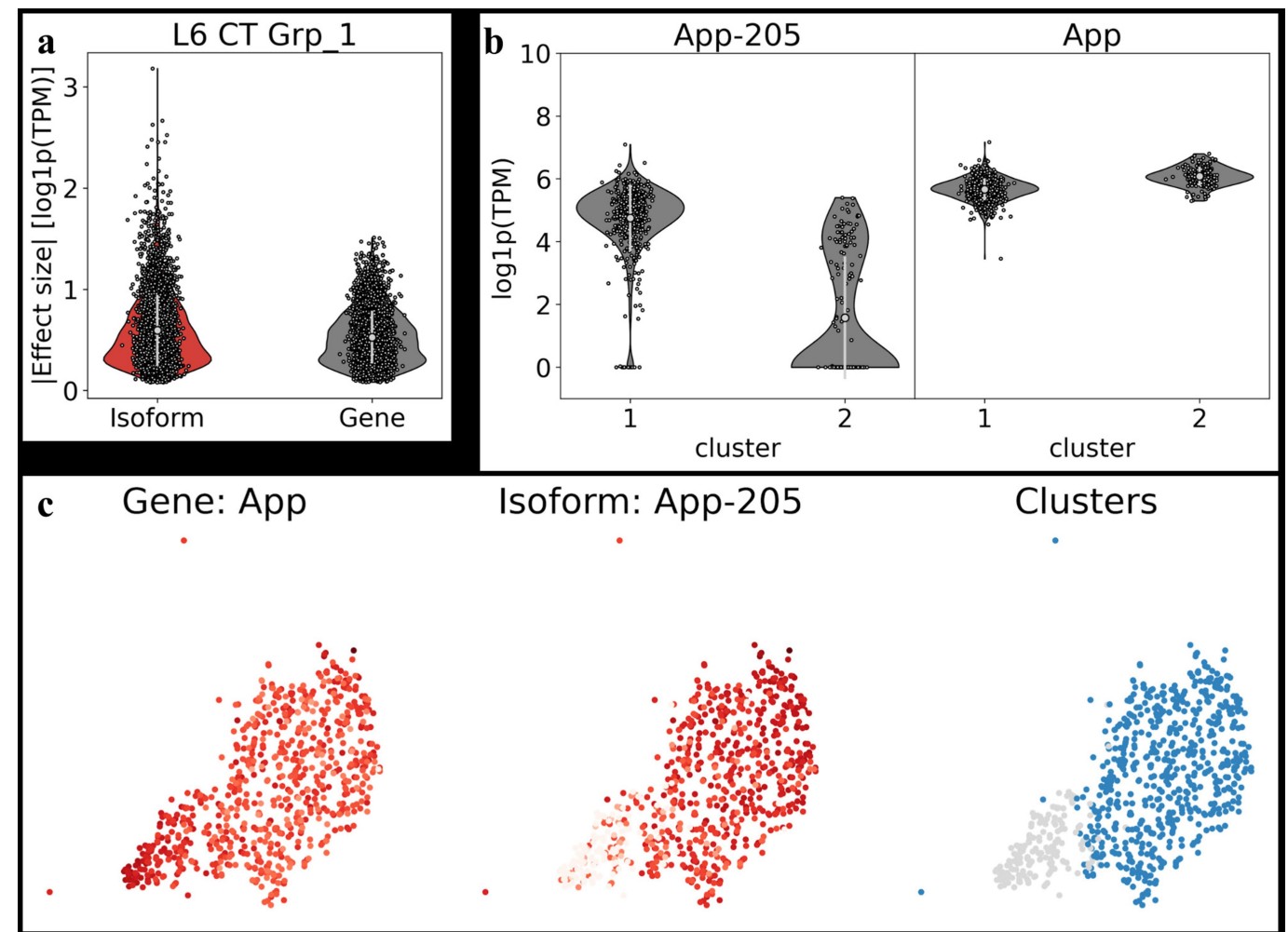

**Extended Data Fig. 5 | Splitting clusters with k-means clustering on isoforms. a**) Effect sizes for isoforms that split the L6 CT Grp_1 cluster into two parts is higher on average than that for genes. **b**) The *App-205* isoform splits the L6 CT Grp_1 cluster into two parts where one part has a higher expression of the isoform whereas both halves have similar *App* gene expression. The white circles within the violin plots represent the mean and the white bars represent ±one standard deviation. **c**) Each point is a cell and is painted by the log1p(TPM) expression of the *App* gene (left) and *App-205* isoform (middle). K-means clustering splits the L6 CT Grp_1 cluster into two distinct halves marked by expression of *App-205*. The white circles within the violin plots represent the mean and the white bars represent ±one standard deviation.

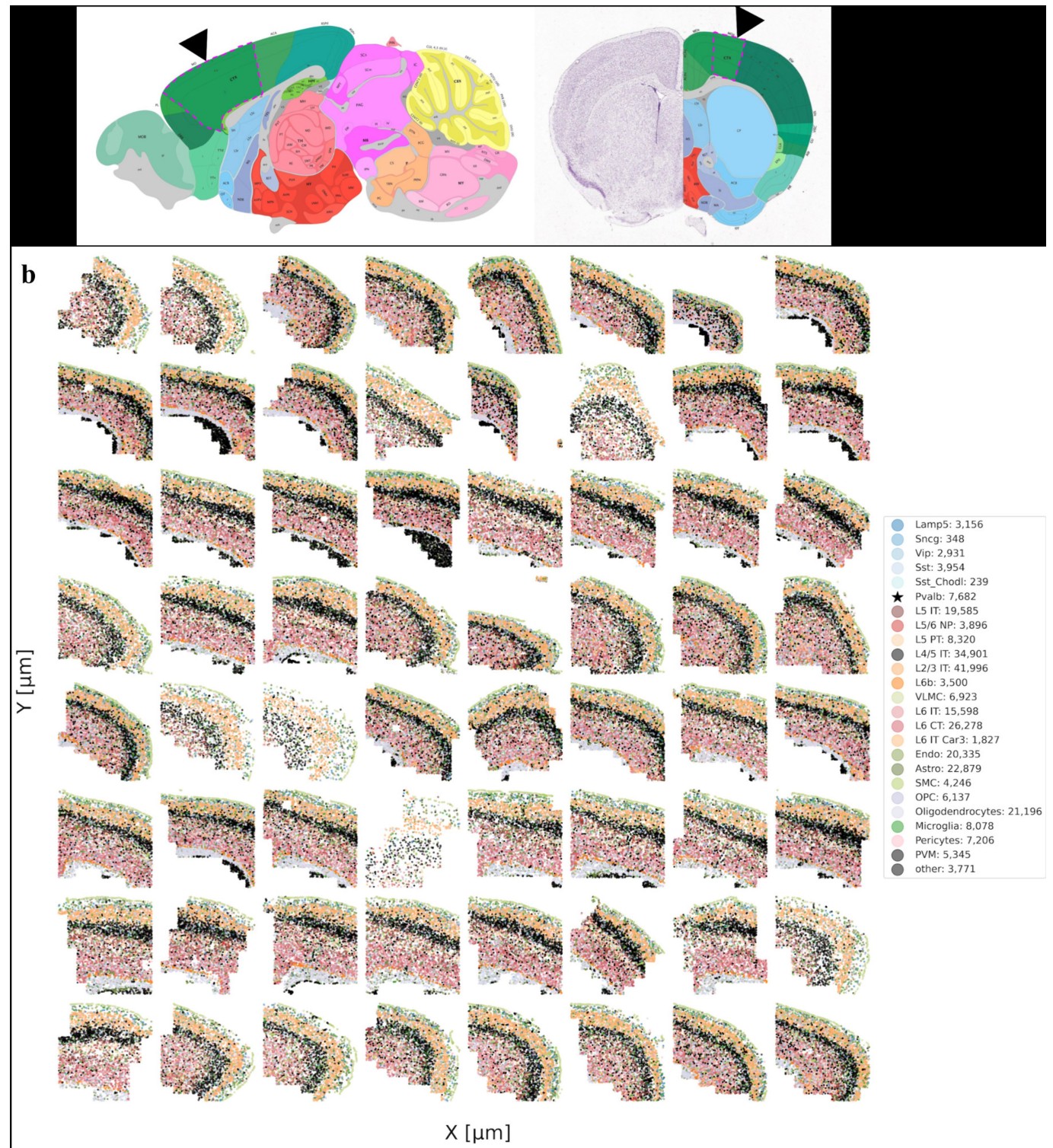

**Extended Data Fig. 6 | Spatial localization of cell types in the mouse primary motor cortex. a**) The location of the mouse primary motor cortex, outlined in pink and pointed to by a black arrow. The sagittal view (left) and coronal view (right) are shown. Image credit: Allen Institute. **b**) Spatial location of cells in all subclasses across 64 slices from the MOp assayed with MERFISH, the *Pvalb* cells are represented by a black star.

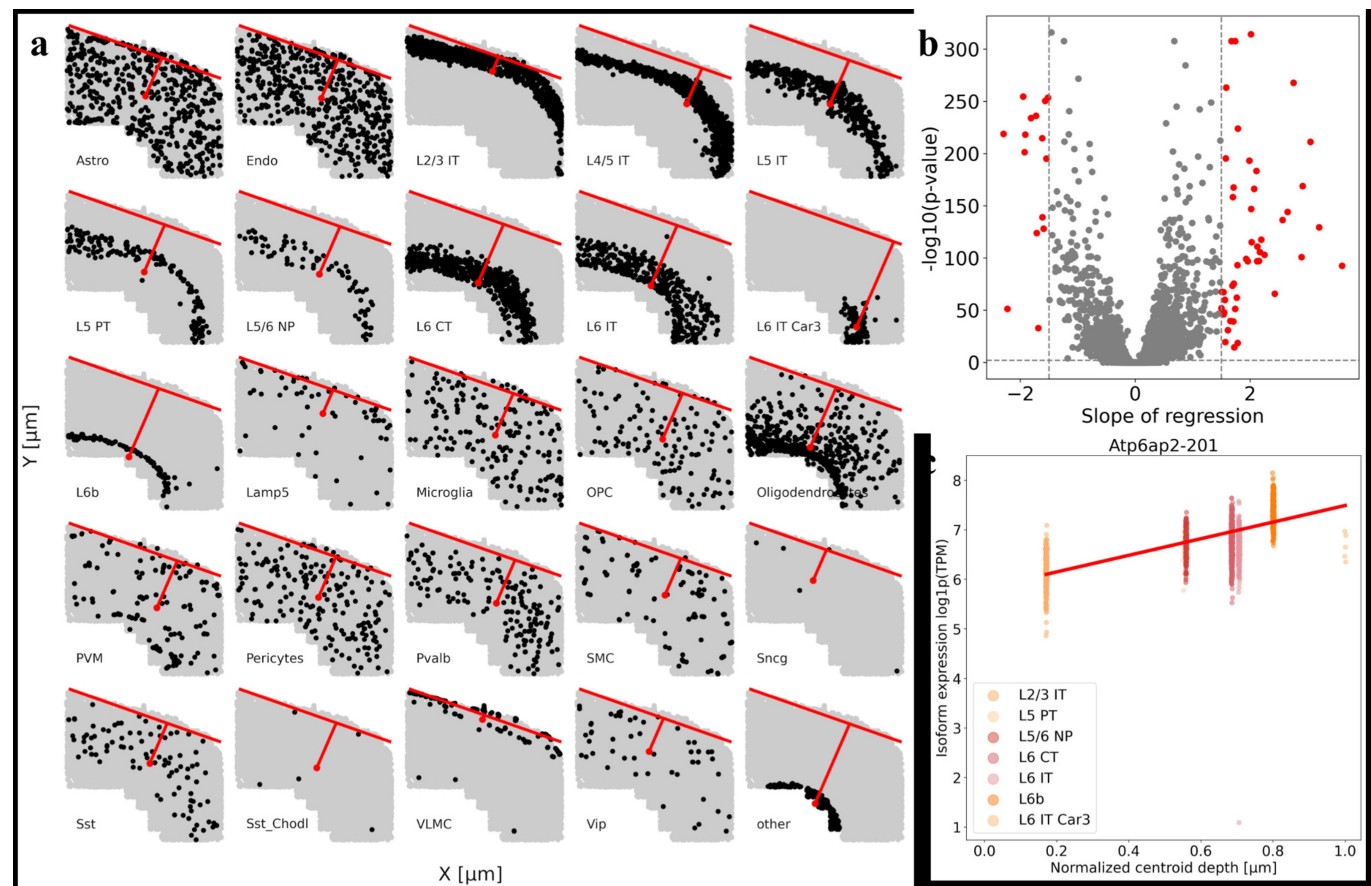

**Extended Data Fig. 7 | Analysis of isoform expression gradients.**
**a**) A representative slice of the mouse primary motor cortex, as assayed by MERFISH, where each dot indicates the position of a cell from the corresponding subclass (black). The red points indicates the position of the centroids of those cells within the colored subclass with a line connecting the centroids to the boundaries of the brain slices; the distance from the centroid to the slice boundary is indicated by the red line. **b**) Volcano plot of the set of isoforms found to be differentially expressed across the depth of the mouse primary motor cortex found using weighted least squares regression. The isoforms with a bonferroni corrected p-value less than 0.01 and regression slope greater than 1.5 are colored red. **c**) An example of an isoform that is colored red in plot (**b**). The expression of *Tubb2a-201* appears to increase across the depth of the motor cortex on average.

**a**

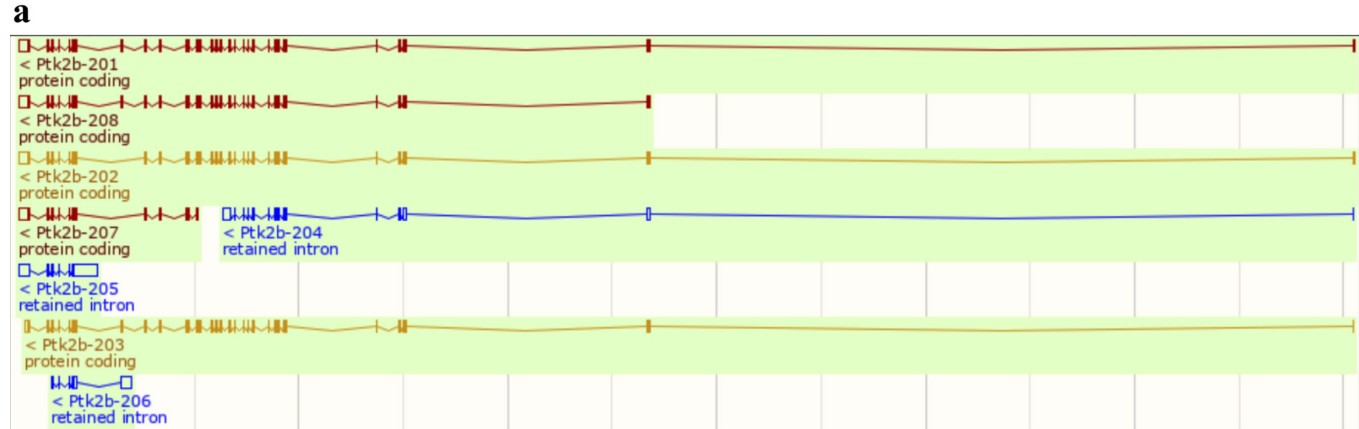

**b**

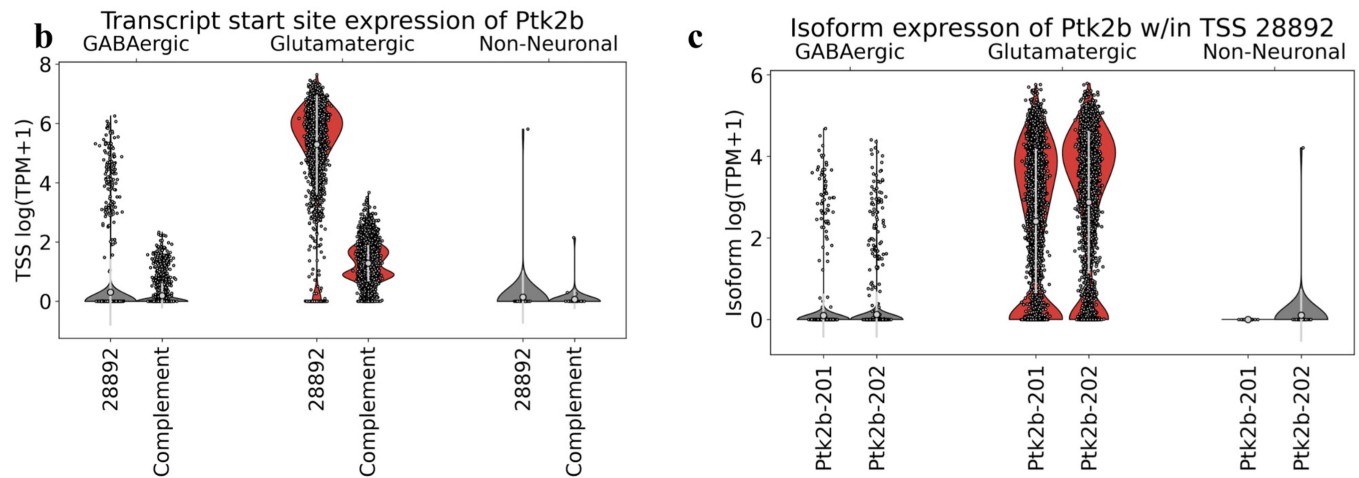

**Extended Data Fig. 8 | Isoform shifts reflecting transcriptional changes. a)** The eight isoforms of the *Ptk2b* gene. The 1st and 3rd isoforms from the top have the same transcription start site at the 5′ end of the transcript. **b)** Expression patterns of groups of transcript sharing the same transcript start site (TSS) from the protein tyrosine kinase 2 (*Ptk2b*) gene. **c)** Expression patterns of isoforms within TSS groups from the *Ptk2b* gene. The white circles within the violin plots represent the mean and the white bars represent ±one standard deviation.

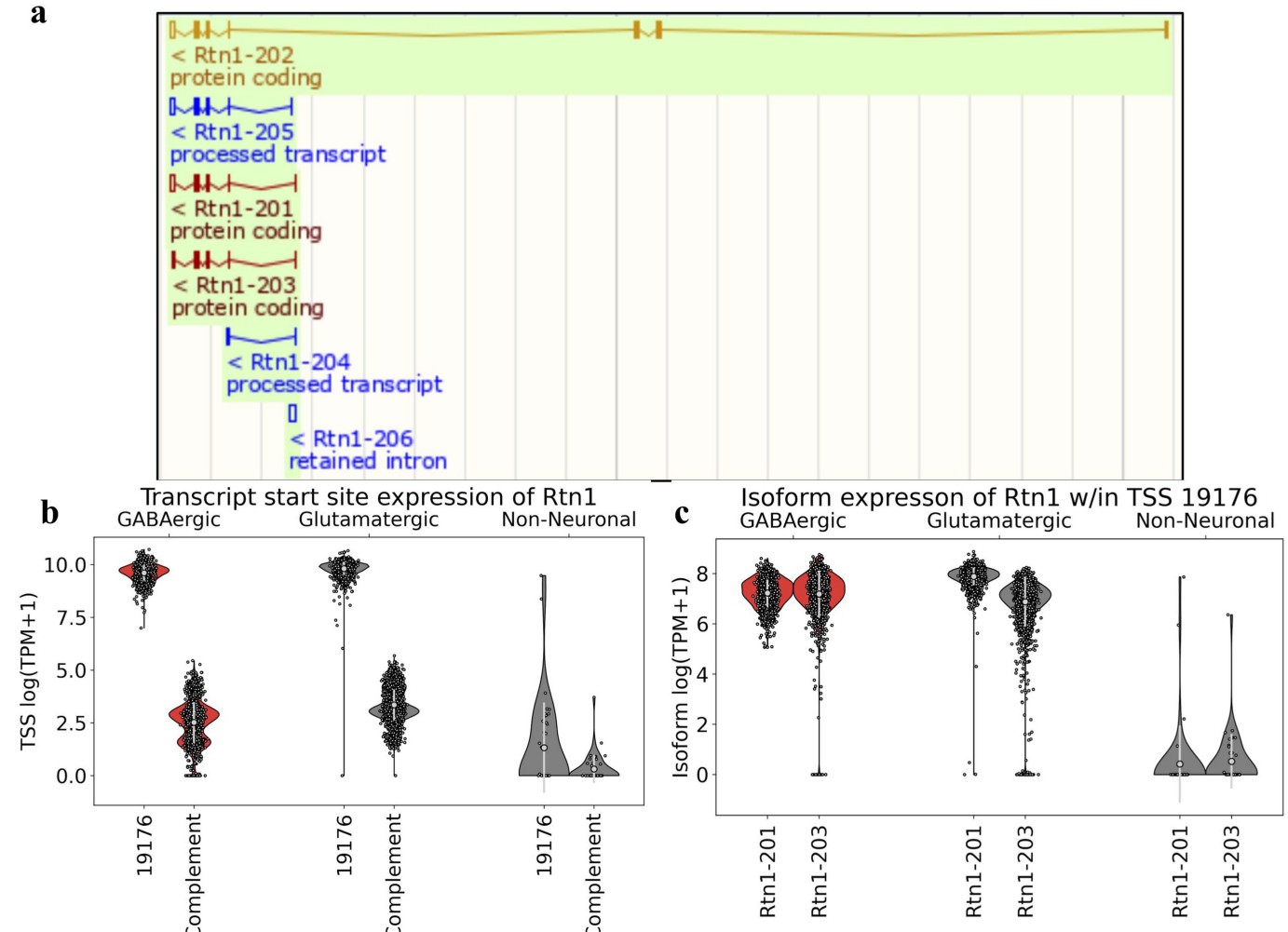

**a**

**b** Transcript start site expression of Rtn1

**c** Isoform expresson of Rtn1 w/in TSS 19176

**Extended Data Fig. 9 | Isoform shifts reflecting post-transcriptional changes. a**) The six isoforms of the *Rtn1* gene. The 3rd and 4th isoforms from the top have the same transcription start site at the 5′ end of the transcript. **b**) Expression patterns of groups of transcript sharing the same TSS from the reticulon 1 (*Rtn1*) gene. **c**) Expression patterns of isoforms within TSS groups from the *Rtn1* gene. The white circles within the violin plots represent the mean and the white bars represent ±one standard deviation.

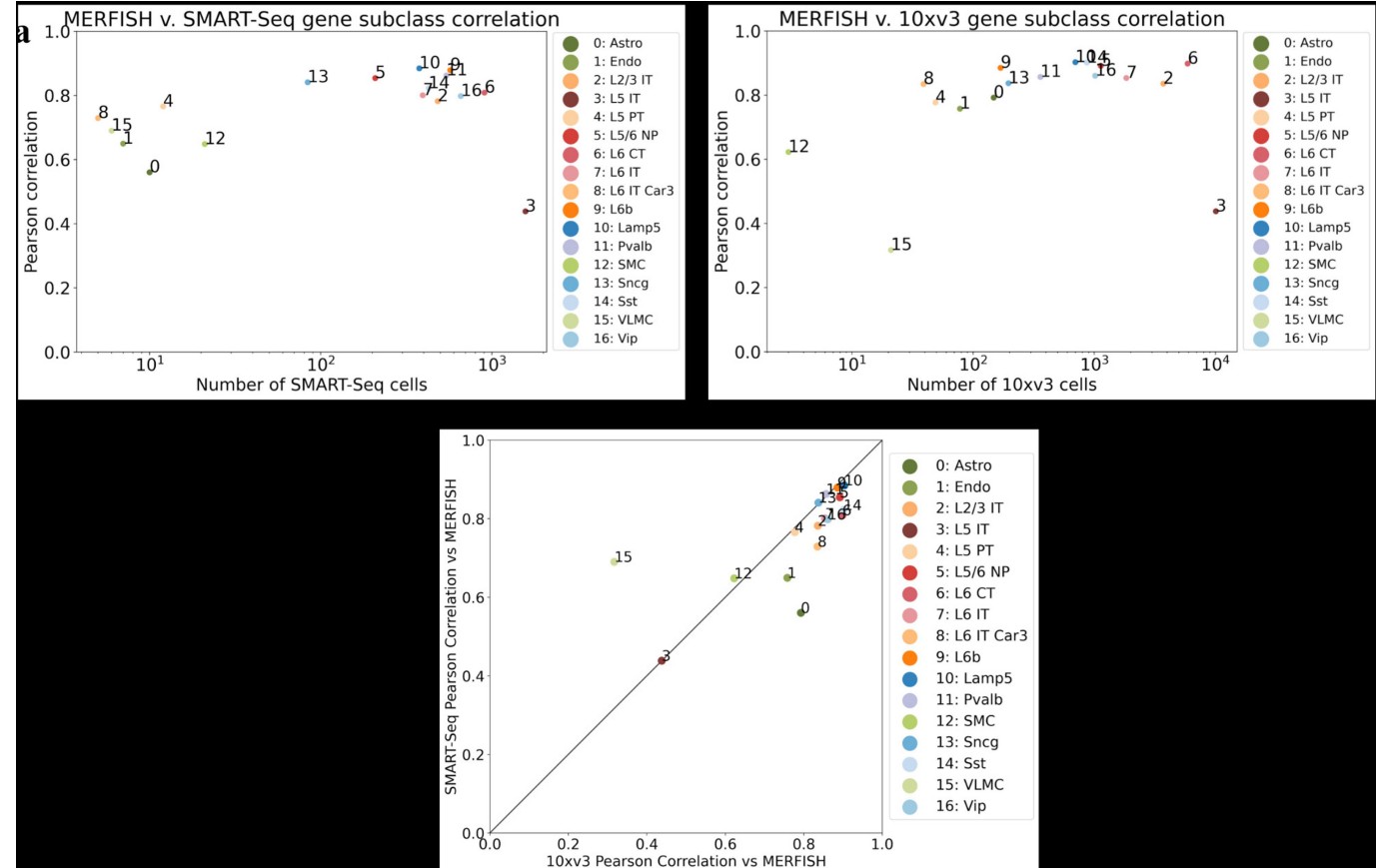

**Extended Data Fig. 10 | Gene level subclass validation with SMART-seq, 10xv3, and MERFISH. a**) Pearson correlation by subclass of the mean gene expression in MERFISH and the mean gene expression in SMART-seq, against the size of the subclass, for all 254 genes in the MERFISH data. **b**) Pearson correlation by subclass of the mean gene expression in MERFISH and the mean gene expression in 10xv3, against the size of the subclass, for all 254 genes in the MERFISH data. **c**) Comparison of gene correlations by cell type between 10xv3 and MERFISH, and SMART-seq and MERFISH computed using the 254 genes assayed in the MERFISH dataset.

# nature research

# Reporting Summary

Nature Research wishes to improve the reproducibility of the work that we publish. This form provides structure for consistency and transparency in reporting. For further information on Nature Research policies, see our Editorial Policies and the Editorial Policy Checklist.

## Statistics

For all statistical analyses, confirm that the following items are present in the figure legend, table legend, main text, or Methods section.

| n/a | Confirmed | |
|---|---|---|
| ☐ | ☒ | The exact sample size (*n*) for each experimental group/condition, given as a discrete number and unit of measurement |
| ☐ | ☒ | A statement on whether measurements were taken from distinct samples or whether the same sample was measured repeatedly |
| ☐ | ☒ | The statistical test(s) used AND whether they are one- or two-sided<br>*Only common tests should be described solely by name; describe more complex techniques in the Methods section.* |
| ☐ | ☒ | A description of all covariates tested |
| ☐ | ☒ | A description of any assumptions or corrections, such as tests of normality and adjustment for multiple comparisons |
| ☐ | ☒ | A full description of the statistical parameters including central tendency (e.g. means) or other basic estimates (e.g. regression coefficient) AND variation (e.g. standard deviation) or associated estimates of uncertainty (e.g. confidence intervals) |
| ☐ | ☒ | For null hypothesis testing, the test statistic (e.g. $F$, $t$, $r$) with confidence intervals, effect sizes, degrees of freedom and $P$ value noted<br>*Give P values as exact values whenever suitable.* |
| ☒ | ☐ | For Bayesian analysis, information on the choice of priors and Markov chain Monte Carlo settings |
| ☒ | ☐ | For hierarchical and complex designs, identification of the appropriate level for tests and full reporting of outcomes |
| ☐ | ☒ | Estimates of effect sizes (e.g. Cohen's *d*, Pearson's *r*), indicating how they were calculated |

*Our web collection on statistics for biologists contains articles on many of the points above.*

## Software and code

Policy information about availability of computer code

| Data collection | No software was used for data collection. |
|---|---|
| Data analysis | Anndata 0.7.1<br>bustools 0.39.4<br>awk (GNU awk) 4.1.4<br>grep (GNU grep) 3.1<br>kallisto 0.46.1<br>kb_python 0.24.4<br>Matplotlib 3.0.3<br>Numpy 1.18.1<br>Pandas 0.25.3<br>Scanpy 1.4.5.post3<br>Scipy 1.4.1<br>sed (GNU sed) 4.4<br>sklearn 0.22.1<br>statsmodels 0.12.1<br>tar (GNU tar) 1.29<br>umap 0.3.10 |

For manuscripts utilizing custom algorithms or software that are central to the research but not yet described in published literature, software must be made available to editors and reviewers. We strongly encourage code deposition in a community repository (e.g. GitHub). See the Nature Research guidelines for submitting code & software for further information.

## Data

Policy information about availability of data

All manuscripts must include a data availability statement. This statement should provide the following information, where applicable:

- Accession codes, unique identifiers, or web links for publicly available datasets
- A list of figures that have associated raw data
- A description of any restrictions on data availability

The single-cell RNA-seq data used in this study was generated as part of the BICCN consortium20. The 10xv3 and SMART-Seq data can be downloaded from http://data.nemoarchive.org/biccn/lab/zeng/transcriptome/scell/. The MERFISH data is available at https://caltech.box.com/shared/static/dzqt6ryytmjbgyai356s1z0phtnsbaol.gz. All cell annotations and cluster labels are available at https://github.com/pachterlab/BYVSTZP_2020/tree/master/reference.

# Field-specific reporting

Please select the one below that is the best fit for your research. If you are not sure, read the appropriate sections before making your selection.

☒ Life sciences  ☐ Behavioural & social sciences  ☐ Ecological, evolutionary & environmental sciences

For a reference copy of the document with all sections, see nature.com/documents/nr-reporting-summary-flat.pdf

# Life sciences study design

All studies must disclose on these points even when the disclosure is negative.

| | |
|---|---|
| Sample size | No explicit calculations were performed to determine sample size. We analyzed 6,160 mouse primary motor cortex cells assayed with SMART-Seq, 280,327 cells assayed with MERFISH, and 94,162 cells assayed with 10x Genomics Chromium v3. We analyzed both male and female mice to understand differences in gene and isoform expression. The sample size for differential expression was set to be such that 90% of cells in a cluster have a non-zero expression of the tested gene. The smallest cluster size contained 7 cells with all cells having non-zero expression of the tested genes. We computed error bars for all tests to ensure that sample sizes were sufficient. |
| Data exclusions | We excluded a subset of the 10xv3 data that was collected on separate dates due to a batch effect. We describe our rationale for exclusion in detail in the manuscript. In short, specific cell-types were enriched for within those batches. |
| Replication | We validated each technology by comparing average cell type expression between MERFISH, SMART-Seq, and 10xv3 at the gene level and at the isoform level for SMART-Seq and 10xv3. These comparisons are replicated and consistent across the three heirarchies: class, subclass, and cluster. We describe the methods in the manuscript. |
| Randomization | All cells that passed quality control were analyzed equally with no sub-sampling therefore there was no requirement for randomization. |
| Blinding | Cell type assignment was not blinded due to practical constraints. |

# Reporting for specific materials, systems and methods

We require information from authors about some types of materials, experimental systems and methods used in many studies. Here, indicate whether each material, system or method listed is relevant to your study. If you are not sure if a list item applies to your research, read the appropriate section before selecting a response.

### Materials & experimental systems

| n/a | Involved in the study |
|---|---|
| ☒ | ☐ Antibodies |
| ☒ | ☐ Eukaryotic cell lines |
| ☒ | ☐ Palaeontology and archaeology |
| ☐ | ☒ Animals and other organisms |
| ☒ | ☐ Human research participants |
| ☒ | ☐ Clinical data |
| ☒ | ☐ Dual use research of concern |

### Methods

| n/a | Involved in the study |
|---|---|
| ☒ | ☐ ChIP-seq |
| ☒ | ☐ Flow cytometry |
| ☒ | ☐ MRI-based neuroimaging |

## Animals and other organisms

Policy information about studies involving animals; ARRIVE guidelines recommended for reporting animal research

| | |
|---|---|
| Laboratory animals | Mice were provided food and water ad libitum and were maintained on a regular 12-h day/night cycle at no more than five adult animals per cage. For this study, we enriched for neurons by using Snap25-IRES2-Cre mice58 (MGI:J:220523) crossed to Ai1459 (MGI: |

J:220523), which were maintained on the C57BL/6J background (RRID:IMSR_JAX:000664). Animals were euthanized at 53–59 days of postnatal age. Tissue was collected from both males and females (scRNA SMART, scRNA 10x v3). (MERFISH, companion paper) Adult C57BL/6 male mice aged 57-63 days were used in this study. Animals were maintained on a 12 hour:12 hour light/dark cycle (2pm-2am dark period) with ad libitum access to food and water.

Wild animals

No wild animals were used in this study.

Field-collected samples

No field-collected samples were used in this study.

Ethics oversight

All procedures were carried out in accordance with Institutional Animal Care and Use Committee protocols at the Allen Institute for Brain Science.

Note that full information on the approval of the study protocol must also be provided in the manuscript.

