## [Peer Review File · Nature]

Manuscript Title: Isoform cell type specificity in the mouse primary motor cortex

Editorial Notes:

Redactions – Third Party Material

Reviewer Comments & Author Rebuttals

Reviewer Reports on the Initial Version:

Referee #1 (Remarks to the Author):

The creation of the isoform marker atlas for the mouse brain is intriguing and convincing. However, the links between the three technologies is not clear, nor is the overall biological significance of the findings. There are some gaps in ruling out technical or computational artifacts that could drive some of the most important results

The authors quantified isoforms using kallisto on SMART-seq data, and t tests were performed for each cluster (vs its complement) and each isoform, as well as each gene. The methodology to find isoforms-specific (but not-gene-specific) clusters is new and interesting. 10x was used to validate the isoform quantifications by SMART-seq by comparing isoforms with unique 3' ends. However it is not clear how or why 10x identifies more clusters than SS2.

Major issues:

- 1) The authors use kallisto to compute isoform abundance. It is not clear how robust the isoform estimates are to the annotation input to kallisto? Could the authors compare different annotations and show robustness of their findings?
- 2) How do the authors control for technical confounders such as artifacts introduced by cell sorting or batch effects?
- 3) The data on which the authors apply t-tests are not normally and or are not approximately normal; this means the results could be biased. How do the authors control for this?
- 4) MERFISH returns spatial information for a limited number of genes (254 in this case) but is not suitable for isoform quantification. If one of these genes is specific to a cluster, and also has a relationship with isoform expression, then we can use the gene level as measured by MERFISH to link the cell with the isoform level (i.e., let's say every cell expressing a high level of Pou3f3 expresses Pvalb isoform 201 (which we'd know based on the SMART-seq data); then when we see a high level of Pou3f3 in MERFISH data, we can assume Pvalb-201 was there too. This appears to be a tenuous way to argue that isoforms are driving differential MERFISH measurements. What if both are driven by technical artifacts?

5) NCA: if this is commonly used for scRNA-seq probably don't need to spend as much time on it; if not then it seems a little randomly thrown into the paper and I'd want to see more of a reason not to use more standard visualizations (PCA). In addition, what's the difference between class, subclass, and cluster? This should be more explicitly defined early on in the paper and results should be presented to show that isoform switching is robust to different clustering methods.

Minor notes:

- Spatial atlas (Extended Data Figure 8): doesn't seem to have specifically spatial information; rather it seems to link clusters with genes/isoforms, which seemingly could be done with just the SMART-seq data
- I believe the "spatial isoform atlas" in Extended Data Figure 8a is only for Pvalb; is that correct? (I

assumed so because of the annotations of isoforms as 201 or 202)

- The naming of the isoform-informed approach at gene count estimation as “valid” in Extended Data Figure 9 seemed a bit presumptuous to me; it seemed like this figure was supposed to argue that it was valid
- Not clear how False positives and false negatives were determined in Extended Data Figure 9
- Please describe more how the work in the paper is a general workflow
- "Our analyses suggest that a workflow consisting of droplet-based single-cell RNA-seq to identify cell

types, then SMART-seq for isoform analysis, and finally spatial RNA-seq with a panel based on isoform-specific markers identified by SMART-seq, would effectively leverage different technologies' strengths" ← this seems to be a broad description of the workflow. This all seems valid, but doesn't necessarily form one cohesive analysis (especially 10x and SMART-seq)

Does the manuscript have flaws which should prohibit its publication?

One of the main aims of this paper, to provide an example of how SMART-seq data can be used in conjunctions with other scRNA-seq technologies, did not seem fully fleshed out in and it is not not totally clear how the analysis was enhanced by the other technologies. Please provide a clear workflow that used them all in conjunction with each other.

Other comments requested:

The methods to synthesize results from multiple single cell sequencing methods are of great interest in the single cell sequencing field currently. This paper is a step towards a workflow that can draw conclusions based on multiple sources of information at the same time. Extrapolation of isoform levels to spatial data is especially relevant, as spatial isoform quantification for multiple genes at the same time is currently difficult. The methodology for identifying isoform markers for clusters seems sound, and a move in the more informative direction of using isoform-level rather than purely gene-level analysis of single cell sequencing data.

No error bars are shown in the manuscript. The t test seems like an appropriate test to identify if the mean isoform level (or gene level) of a cluster was different from its complement (though I don't believe it's addressed whether the data would be expected to follow a normal distribution). P values were corrected using Bonferroni correction.

Referee #2 (Remarks to the Author):

This manuscript explores the use of SMART-seq single-cell RNA-seq data can be used to analyze isoform expression in single cells, analyzing over 6000 cells from the mouse primary motor cortex, using SMART-seq v4 and also comparisons to larger numbers of cells analyzed using the 10x Genomics Chromium and MERFISH approaches. The conclusions reached are that specific mRNA isoforms can often be used to distinguish cell types, even when no difference in expression is apparent at the gene level. Several clear examples of this phenomenon are shown. From a biological point of view, this result is not terribly surprising as alternative splicing is known to vary substantially during cellular differentiation and between various mammalian cell and tissue types from previous studies using a variety of approaches (e.g., PMID 17606642, 28673540, 30028642). The specific analysis of isoform-specific usage between glutamatergic and GABAergic neurons is interesting but largely confirmatory of the previous work on differential single-cell exon usage between the same types of neurons in the primary visual cortex that inspired this analysis (<https://www.nature.com/articles/nn.4216>). The novelty is therefore more of a technical nature, in the study of isoform distributions at single cell resolution and their application as cell type markers.

A challenge in the use of scRNA-seq to assess isoform abundance is the limited capture efficiency of mRNAs, which results in the problem of gene dropouts (failing to capture any reads from a gene that is expressed), which is even more serious at the level of mRNA isoforms which are expressed at levels that may be a small or large fraction of the expression of the corresponding gene, and which require

reads to cross the particular splice junctions that distinguish them from related isoforms. More explicit attention to issues of capture efficiency and coverage (see below) would make the results more convincing.

Cross-platform comparisons are also used to assess the strengths and weaknesses of different approaches, a worthwhile technical contribution. The manuscript is written clearly and the authors have done a commendable job in ensuring the reproducibility of their work by providing clear methods and links to code, etc. However, what is almost completely lacking from the manuscript is some sense of the biological importance of the descriptions of differential isoform usage across cell types and of the inferences the authors make – the paper feels descriptive and lacking in context. The main conclusions relate to which combinations of methodologies are best for scRNA-seq analyses, and which isoforms can be used as markers of particular cell types, but these conclusions are likely to be of interest primarily to specialists. No attempt is made to draw biological conclusions that might be of interest to a broader audience.

Specific issues:

1. Recent work has challenged the applicability of most scRNA-seq datasets to reliably estimate isoform abundance from all but the most highly expressed genes (<https://www.biorxiv.org/content/10.1101/2019.12.19.883256v1>), showing by simulation that presence of multiple isoforms in a single cell is often obscured by limited coverage resulting from insufficient mRNA capture efficiency in scRNA-seq library preps. Further, this study shows that isoform abundance can only reliably be estimated from scRNA-seq data by modeling coverage of each gene based on estimates of mRNA capture efficiency and expression level to assure sufficient depth of coverage. More attention should be given to these issues to convincingly show that the genes and datasets analyzed have sufficient coverage to reliably estimate isoform abundance.

2. In the section accompanying Fig 3 the authors use single-cell data to note that of the two isoforms of the Pvalb gene, only one (Pvalb-201) is expressed in the MOp region of the brain, and then extrapolate based on cell-type markers to map the isoform expression patterns onto spatial expression data. Given that only one of the two isoforms is expressed at a detectable level, the extrapolation reduces to the inference that any cell which expresses Pvalb expresses the Pvalb-201 isoform. That is, the authors are only mapping the spatial patterns of Pvalb isoform expression by virtue of the fact that it is the only isoform detectably expressed. This seems like a reasonable extrapolation, but it is not clear that it provides any clear advantage in terms of cell type classification over the use of the Pvalb gene as a marker, or other desirable outcome. Can the authors explain why this extrapolation adds something important, or provide a more compelling example of where an important biological inference is enabled by this sort of extrapolation?

3. The figures are somewhat difficult to follow and need better legends. For example:

- a. In Fig1e and e what do the dotted lines and circles represent? This should be explained in the legend.
- b. Figure 2 and Ext Fig 8 are somewhat visually overwhelming. In interpreting the 289 distributions displayed as violin plots, what should the reader focus on? Most readers will focus on the centers of the distributions, or just the presence or absence of expression. Further, the variable y-axes and that the bulk of many distributions (and their means) are cut off by the axis limits makes this plot difficult to read. If the mean expression level is the most important piece of information, a heat map might be easier for readers to digest.
- c. The legend to Figure 3a appears to apply to 3c, while the legends to 3b and 3c appear to apply to 3a and 3b.
- d. In Figure 4c, the isoforms should be labeled for clarity.

Author Rebuttals to Initial Comments:

Referee #1 (Remarks to the Author):

The creation of the isoform marker atlas for the mouse brain is intriguing and convincing. However, the links between the three technologies is not clear, nor is the overall biological significance of the findings. There are some gaps in ruling out technical or computational artifacts that could drive some of the most important results

The authors quantified isoforms using kallisto on SMART-seq data, and t tests were performed for each cluster (vs its complement) and each isoform, as well as each gene. The methodology to find isoforms-specific (but not-gene-specific) clusters is new and interesting. 10x was used to validate the isoform quantifications by SMART-seq by comparing isoforms with unique 3' ends. However it is not clear how or why 10x identifies more clusters than SS2.

Major issues:

1) The authors use kallisto to compute isoform abundance. It is not clear how robust the isoform estimates are to the different annotations and show robustness of their findings?

To demonstrate the robustness of our findings to the choice of reference we re-quantified all 15,229,289,826 SMARTSeq reads against the GENCODE M25 mouse transcriptome reference. The estimated isoform abundances are consistent between the BICCN reference annotation and the Gencode reference annotation, with a mean Pearson correlation of 0.965 across all 107,639 common isoforms. We have added this result to the main text and have included the distribution of correlation across isoforms as Extended Figure 6. These results show that our quantifications are robust to annotation.

We have also expended considerable effort with this revision to update our Jupyter notebooks (see https://github.com/pachterlab/BYVSTZP_2020/tree/master/analysis/notebooks) so as to ensure that the entire analysis is reproducible, and all of our results can therefore be easily updated with respect to new annotations in the future. Running our entire workflow and quantifying isoform abundances with the EM algorithm on the ~15 billion read dataset we have processed takes approximately 6 hours using 16 threads on our server. This is thanks to the fast pre-processing of the data with the kallisto program.

2) How do the authors control for technical confounders such as artifacts introduced by cell sorting or batch effects?

Thank you for raising this point. First, upon reviewing our manuscript in light of your question we realized we did not effectively explain that while we leverage the 10xv3 data for validation purposes and for cluster identification, our isoform analysis of the SMART-seq data is independent of the 10xv3 data. In other words, our analysis is sensitive to potential batch effects between the SMART-seq and 10xv3 datasets only with respect to the subclass and cluster comparisons. We have added a Methods description

to clarify how the SMART-seq labels were obtained and to highlight that we did not analyze the data in a manner that is affected by the 10xv3 data. It is true that cell sorting was performed prior to the single-cell RNA-seq, and while in principle this could result in a bias in cells assayed, the detailed analyses and validation of the MOp transcriptomics paper co-submitted with our manuscript to Nature (BRAIN Initiative Cell Census Network (BICCN) et al. 2020) (Tasic et al. 2018) (Yao et al. 2020) provides ample evidence, via known cell types, gene markers etc., that the cells assayed comprehensively and uniformly cover the MOp.

To determine if there were significant batch effects that may have affected the 10x-SMART-Seq comparison we began by looking at sex as one of the possible confounders, since we noticed that the ratio of Male:Female cells was not the same in the two datasets. In the 10x data we observed that sex resulted in separation of all clusters in the t-SNE indicating a batch effect:

This disturbing figure worried us. Moreover, since not all clusters were segregated completely based on sex we hypothesized that there was possibly an additional confounding batch effect. After examining the sample metadata more carefully, we observed that the data was collected in three batches on three different dates (November 29, 2018, December 7, 2018 and April 26, 2019), and that sex was confounded with these batches. When painting cells based on their assay date we observed that almost all clusters were entirely segregated by batch, thus indicating a significant batch effect in the 10x data:

To mitigate this problem we restricted the analysis to the third 10xv3 dataset that contained both male and female cells and covered all the cell types. We have included this visualization as a new Extended Data Figure 8a and 4a and have also notified the transcriptomics group within the BICCN consortium of the presence and significance of this batch effect in the data.

The restricted dataset passes our quality control (inset, bottom right of Extended Data Figure 8a), also reproduced below. Now the cells are mixed with respect to sex in every subclass:

We therefore recomputed our results with respect to it (new Extended Data Figure 1). Our analysis is now based on 26,870 10x cells. We note that the previous low correlation we saw for the L5 IT subclass was directly due to this batch effect, and while we had

discarded analysis of it earlier assuming there was some underlying problem with the data, we now know what the problem is, and we thank you for leading us to this discovery.

The SMART-seq data, which is collected in plates, has the property that effectively every cell is a “batch”. We did not notice significant issues associated with the metadata we examined. After restricting the 10xv3 data to one batch, we see high correlations in quantifications between all three technologies we analyzed (10xv3, MERFISH and SMART-seq), suggesting that there was high reproducibility and no additional significant technical confounding variables in any single technology. The MERFISH technology has been separately validated (Chen et al. 2015; Zhang et al. 2020), but we also searched for batch effect in that dataset and didn’t see anything out of the ordinary (see new Extended Data Figure 4b).

There is one additional potential source of batch effect that could have affected our corrected results, and that is that batch effect may be present in the four different 10xv3 libraries prepared on the same day, which we did combine for our analysis. Your question prompted us to check for batch effect between the lanes sequenced on that date. We performed pairwise comparison of gene counts for each of the 4 10xv3 batches and found the Pearson correlation to be very high for all pairs, with a mean of 0.9979 (see Figure below). We note that the correlation was just under 1.0, but is reported as 1.0 in the plots due to rounding.

We also looked at the distribution of batch labels across clusters and subclasses, and found that the observed fraction of cells per batch in each cluster was almost exactly the expected fraction of cells per batch assuming uniform mixing (see Figure below):

These results confirm there is minimal batch effect in the filtered 10xv3 data we have now used to validate the SMART-seq and that our analysis of them together is valid. We have added the statements about the validity of our comparisons as noted above to the paper but have not included the two figures above because we don't believe they provide much value to readers beyond the summary statistics.

3) The data on which the authors apply t-tests are not normally and or are not approximately normal; this means the results could be biased. How do the authors control for this?

We chose to use the t-test since the t-test was found to exhibit low bias as one of the top performing methods of 36 differential expression methods (ranked by 14 different criteria) studied by (Figure 5, Sonesson and Robinson 2018). Our straightforward experimental design makes the test suitable for use according to Sonesson et al. Additionally, we filtered out lowly expressed genes since the Sonesson & Robinson analysis on real and synthetic datasets found that results were significantly improved after filtering, namely p-values were uniformly distributed under the null hypothesis after filtering for lowly expressed genes.

4) MERFISH returns spatial information for a limited number of genes (254 in this case) but is not suitable for isoform quantification. If one of these genes is specific to a cluster, and also has a relationship with isoform expression, then we can use the gene level as measured by MERFISH to link the cell with the isoform level (i.e., let's say every cell expressing a high level of Pou3f3 expresses Pvalb isoform 201 (which we'd know based on the SMART-seq data); then when we see a high level of Pou3f3 in MERFISH data, we can assume Pvalb-201 was there too. This appears to be a tenuous way to argue that isoforms are driving differential MERFISH measurements. What if both are driven by technical artifacts?

We agree that MERFISH is not suitable for isoform quantification and that in order to infer spatial isoform expression the isoform quantification and spatial gene expression must not be driven by technical artifacts. Inference of spatially resolved isoform markers is only performed if the gene marking a specific cell type in the spatial data has one of its isoforms marking the same cell type in the isoform data. This one-to-one relationship ensures that spatial isoform inference is specific to genes/isoforms that are localized to a cell type instead inferring relationships between isoforms and genes that mark cell types.

Additionally, given the numerous studies validating SmartSeq for isoform quantification and MERFISH for spatial gene expression we do not think it is the case that inferring spatial isoform expression for a specific isoform is a tenuous argument. SmartSeq has been studied and validated extensively in the context of identifying isoforms:

- (Ramsköld et al. 2012) Which first introduced the Smart-Seq protocol found that Smart-Seq “...has improved read coverage across transcripts, which significantly enhances detailed analyses of alternative transcript isoforms and identification of SNPs.”
- (Picelli et al. 2014) Introduced SMART-Seq2 which improved on the original SMART-Seq protocol in numerous ways. One in particular is the uniformity of read coverage across transcripts for all genes. Supplementary Figure 9a is reproduced below:

[Redacted]

- (Seirup et al. 2020) Find that Smart-seq’s higher sensitivity and read-depth allow for analysis of lower expressed genes and isoforms in their biological system, identifying isoforms with distinct expression in cases where 10x and MarsSeq could not.

Similarly, MERFISH has been studied and validated extensively on many panels of genes in other biological systems as well as in the mouse primary motor cortex as a BICCN

companion paper (Chen et al. 2015; Zhang et al. 2020). We use these complementary methods for measuring RNA expression to establish a link between gene space, isoform space, and physical space, and in doing so leverage the technologies in a way where the whole is greater than the sum of the parts. We have clarified this in a new figure we have included in the manuscript (Figure 1).

5) NCA: if this is commonly used for scRNA-seq probably don't need to spend as much time on it; if not then it seems a little randomly thrown into the paper and I'd want to see more of a reason not to use more standard visualizations (PCA). In addition, what's the difference between class, subclass, and cluster? This should be more explicitly defined early on in the paper and results should be presented to show that isoform switching is robust to different clustering methods.

Standard visualization techniques developed for single-cell RNA-sequencing typically use a combination of a linear dimensionality reduction such as PCA followed by a non-linear dimensionality reduction such as t-SNE. The idea is that PCA will find a subspace that maximizes variance in the data, and t-SNE will faithfully project that subspace into two dimensions all while preserving the local/global structure of the cells.

While NCA is an established method in the field of machine learning, to our knowledge it has not been coupled with t-SNE before as we did in this manuscript. Interestingly NCA was published by the developer of t-SNE, see (Goldberger et al. 2004), but the two have not been linked previously. The reason we are doing so is not by choice, but by necessity: unlike standard single-cell RNA-seq studies, we had a special situation where we needed to visualize the SMART-Seq data with predefined cluster labels produced via a joint analysis with many other data types that had already been undertaken by the BICCN. This was essential in order to ensure that our analysis was concordant with the BICCN flagship manuscript. Thus we had a unique challenge; typically, in single-cell RNA seq, data is usually clustered de-novo and then visualized. We realized as a result of your question that we had not explained this well in the manuscript, and have therefore now added discussion of this in the main text, and have also added the rationale to the methods section to clarify this important point in the paper:

“We first sought to visualize our SMART-Seq data using gene derived cluster labels from the BICCN analysis (see Methods). Rather than layering cluster labels on cells mapped to 2-D with an unsupervised dimensionality reduction technique such as t-SNE¹⁸ or UMAP¹⁹, we utilized a supervised learning approach to project cells so that they are best separated according to BICCN consortium (Yao et al. 2020) annotations using neighborhood component analysis (NCA). ”

Regarding the terms class/subclass/cluster, thank you for pointing out that we forgot to define them. This was a major oversight, and we have fixed it by adding the following to the manuscript:

“The clustering method generates three hierarchies of cells: classes, subclasses, and clusters. The SMART-Seq data has 2 major classes (Glutamatergic, GABAergic), 18 subclasses that subdivide the classes, and 62 clusters that subdivide the subclasses.”

Minor notes:

- Spatial atlas (Extended Data Figure 8): doesn't seem to have specifically spatial information; rather it seems to link clusters with genes/isoforms, which seemingly could be done with just the SMART-seq data

Thank you for pointing this out, the figure did not explicitly contain spatial information and relied on the knowledge that each subclass has a physical location in the MERFISH data. We regret omitting this. We have added this to Figure 3 and have also added plots to the left of each subclass indicating their physical location within a slice of the mouse primary motor cortex (see also Supplementary Table 9) . We have also clarified and extended the caption per your suggestion. In this figure we are showing genes that are differential in the MERFISH data and the underlying isoforms that are differential in that cluster in the SMART-Seq data. This is a more detailed look of spatial isoform inference. If one just knew the differential isoforms for that cluster then the positional coordinate of the cell would be lost. There are methods (Lebrigand et al. 2020) being developed to perform spatial isoform sequencing but these suffer from the limited number of distinct isoforms that can be detected. The novelty of our approach combines multiple methods of measuring RNA inside cells to provide insights that no single method could generate on its own; full length isoform data overcomes the lack of full isoform resolution in spatial methods.

- I believe the “spatial isoform atlas” in Extended Data Figure 8a is only for Pvalb; is that correct? (I assumed so because of the annotations of isoforms as 201 or 202)

We have clarified the caption for Extended Data Figure 8 (now Figure 3) and have included additional clarifying plots. The isoform atlas is showing the physical location of sets of cells (black points) belonging to a subclass (left scatter plots) in a slice of the mouse primary motor cortex. The violin plots are showing expression of marker isoforms from the SMART-Seq data (along the diagonal) where the MERFISH marker genes (columns) are also differential for each of the subclasses (rows). Spatial isoform inference links isoform expression from the SMART-Seq data with physical location of the cells expressing that isoform from the MERFISH data.

- The naming of the isoform-informed approach at gene count estimation as “valid” in Extended Data Figure 9 seemed a bit presumptuous to me; it seemed like this figure was supposed to argue that it was valid

Thanks for your comment. You are right that the word “valid” was inappropriate. We have added a discussion point in the methods along with supporting citations to explain what we mean. This analysis attempts to show that normalizing full length isoform quantifications is important in determining marker genes. Since reads can come from anywhere in the transcriptome it is likely that read counts for longer isoforms are enriched. Therefore it is important to normalize isoform quantification by length prior to downstream analysis. This is supported by extensive analysis in bulk RNA-seq (Jiang and Wong 2009; Trapnell et al. 2013). We have therefore renamed the gene count estimate to “EM” estimate to reflect the method by which transcript abundances are quantified, rather than calling them “valid”, which is indeed a poor descriptor.

- Not clear how False positives and false negatives were determined in Extended Data Figure 9

This was poorly worded and we have changed the language. Since we have no “ground truth” it is impossible to claim that a gene is a false positive or a false negative marker gene when comparing the EM gene quantifications vs the naive strategy. We now simply note that results can be very different with and without the EM algorithm, and we rely on the extensive prior literature establishing much better accuracy with the EM algorithm for the reader to deduce the practical implications of utilizing the naive counting method. We have added a few sentences to the Methods section elaborating this point. The relevant figure is now Extended Data Fig. 15.

- Please describe more how the work in the paper is a general workflow

We have created a new figure (Figure 1) and have added a paragraph in the discussion section explaining the general workflow. Briefly, we propose using UMI-based gene tagging methods for high cell capture at low depth in order to identify rare cell types, full-length isoform methods for isoform quantification on the cells of interest, and spatial methods for determining isoform localization or spatial methods that use isoform quantification to develop isoform panels for cell types of interest. This multi-platform approach measures RNA in three different ways with each way providing a valuable but orthogonal piece of supporting information for studying the MOp. The result of our approach is a “whole is greater than the sum of its parts”, illustrated graphically in Figure 1.

- "Our analyses suggest that a workflow consisting of droplet-based single-cell RNA-seq to identify cell types, then SMART-seq for isoform analysis, and finally spatial RNA-seq with a panel based on isoform-specific markers identified by SMART-seq, would effectively leverage different technologies' strengths" ← this seems to be a broad description of the workflow. This all seems valid, but doesn't necessarily form one cohesive analysis (especially 10x and SMART-seq)

We agree, and appreciate that we did not make it clear exactly what the workflow entailed. As mentioned above we created a new descriptive figure (Figure 1 in the main manuscript) that demonstrates how we leverage three distinct ways of measuring RNA content of cells to infer spatial isoform information. The workflow consists of using primarily gene tagging technologies such as 10x Chromium to perform cell type identification, isoform sequencing with technologies such as SMARTSeq to identify isoform markers in cell types, and spatial RNA capture with technologies such as MERFISH to spatially place cell-type isoform markers. We have also added an extended discussion point to address how these three methods are used. We note that our transparent code, fully reproducing our workflow in a series of 56 notebooks (see https://github.com/pachterlab/BYVSTZP_2020/tree/master/analysis/notebooks), is in and of itself a detailed Methods section, leaving nothing to the imagination.

Does the manuscript have flaws which should prohibit its publication?

One of the main aims of this paper, to provide an example of how SMART-seq data can be used in conjunctions with other scRNA-seq technologies, did not seem fully fleshed out in and it is not not totally clear how the analysis was enhanced by the other technologies. Please provide a clear workflow that used them all in conjunction with each other.

As mentioned above we have created a new Figure and have added an extended discussion in the results section on how the three methods are used in conjunction and how each method enhances the other. We have used the analogy of positive epistasis from population genetics where “the whole is greater than the sum of the parts”. We feel this is an apt descriptor for what we are doing. We are using different measurements of the same quantity, each with distinct advantages and disadvantages, to ultimately improve the precision with which we understand the RNA content of the MOp.

Other comments requested:

The methods to synthesize results from multiple single cell sequencing methods are of great interest in the single cell sequencing field currently. This paper is a step towards a workflow that can draw conclusions based on multiple sources of information at the same time. Extrapolation of isoform levels to spatial data is especially relevant, as spatial isoform quantification for multiple genes at the same time is currently difficult. The methodology for identifying isoform markers for clusters seems sound, and a move in the more informative direction of using isoform-level rather than purely gene-level analysis of single cell sequencing data.

No error bars are shown in the manuscript. The t test seems like an appropriate test to identify if the mean isoform level (or gene level) of a cluster was different from its complement (though I don't believe it's addressed whether the data would be expected to follow a normal distribution). P values were corrected using Bonferroni correction.

Thank you for pointing out the lack of error bars. We have added them to our figures indicating +/- one standard deviation of the mean and have noted this in the Methods section (added to Figure 2,). As we mentioned above in the response to Reviewer 1, the use of the t-test was justified by (Soneson and Robinson 2018). In their analysis the t-test was one of the top performing differential expression methods among 36 different methods.

Referee #2 (Remarks to the Author):

This manuscript explores the use of SMART-seq single-cell RNA-seq data can be used to analyze isoform expression in single cells, analyzing over 6000 cells from the mouse primary motor cortex, using SMART-seq v4 and also comparisons to larger numbers of cells analyzed using the 10x Genomics Chromium and MERFISH approaches. The conclusions reached are that specific mRNA isoforms can often be used to distinguish cell types, even when no difference in expression is apparent at the gene level. Several clear examples of this phenomenon are shown. From a biological point of view, this result is not terribly surprising as alternative splicing is known to vary substantially during cellular differentiation and between various mammalian cell and tissue types from previous studies using a variety of approaches (e.g., PMID 17606642, 28673540, 30028642). The specific analysis of isoform-specific usage between glutamatergic and GABAergic neurons is interesting but largely confirmatory of the previous work on differential single-cell exon usage between the same types of neurons in the primary visual cortex that inspired this analysis (<https://www.nature.com/articles/nn.4216>). The novelty is therefore more of a technical nature, in the study of isoform distributions at single cell resolution and their application as cell type markers.

While our manuscript does present a novel technical framework for studying isoforms at single-cell resolution, there are several additional contributions we feel are important:

- **We present, for the first time, a catalog of isoform markers at different hierarchies of the cell type classification of the BICCN (Yao et al. 2020; BRAIN Initiative Cell Census Network (BICCN) et al. 2020; Zhang et al. 2020). Our work is a companion to the numerous other BICCN projects on the primary motor cortex, and our manuscript is the (only) one presenting the isoform resolved atlas. We believe that this resource will be of value to the community.**
- **While it is true that several of our isoform markers are well known, a result which provides validation of our novel technical framework, our catalog contains many additional isoforms that have not been studied in detail, and these are prime candidates for follow-up studies. As we explain below, there are now experimental techniques for such isoform-level studies (e.g., (Thomas et al. 2020)), an advance in the field that goes hand-in-hand with our work.**
- **Our workflow and framework for isoform analysis is built on reproducible, transparent, and efficient code that will provide researchers direct access to all types of analyses with BICCN transcriptomic data, not just at the isoform-level but also at the gene-level. This is another key resource contribution of our paper. As we remarked in a response to Reviewer #1, all of our results are reproducible in about 7 hours making possible updates to the BICCN catalogs when annotations are updated and improved.**
- **Following up on other remarks, also from the other reviewer, we highlight several interesting biological findings. We acknowledge that detailed follow-up experiments on some of our interesting findings are beyond the scope of this paper.**

To summarize: Overall we identify 5,658 isoforms from 3,132 genes that mark the major classes and 7,588 isoforms belonging to 4,171 genes within the glutamatergic class and 4,359 isoforms belonging to 2,614 genes within the GABAergic class marking the subclasses. At the cluster level for the 48 clusters passing filter 3,171 isoforms belonging to 2,461 genes mark the cluster. Thanks to your remarks and that of the other reviewer, we have gone to considerable lengths to organize this catalog better in a new Supplementary Table (Supplementary Table 9).

We note that importantly, we find isoform markers when the overlying gene does not mark the cell type. We show that 398 isoforms belonging to 310 genes mark the classes and 654 isoforms from 550 genes within the glutamatergic class and 381 isoforms from 332 genes within the GABAergic mark the subclasses even when the overlying gene is constant. This highlights the importance of isoform quantification with full length RNA sequencing. Additionally our methodology for identifying spatially-resolved isoform markers yields 16 subclasses with their spatial location and associated isoform markers, and crucially, we highlight isoform markers that split clusters indicating possible new cell types, a result that has bearing on numerous other single-cell RNA-seq studies with both SMART-seq and 10x Chromium data (there are many such studies, see (Svensson et al. 2020)). We also describe isoforms that we find to be differential between males and females, and we now highlight isoform markers that provide exciting glimpses into the varied biology occurring at the isoform level.

In response to the question about biological discovery, we introduce a method for studying isoform variability across the depth of the mouse primary motor cortex by using MERFISH data to screen for gradients across subclass depth and SMART-Seq data to determine isoform expression across the depth. We have added a new Extended Data Figure 12 and have added a discussion point on this. While we find many isoforms that exhibit a significant change associating with depth, none of the isoforms passing our filters exhibit a monotonic change with respect to the mean. This suggests that non-linear models may be better suited to study isoform variability across the depth of the mouse primary motor cortex (Rash and Grove 2006; Sansom and Livesey 2009).

As mentioned above, when curating the isoform atlas we identified numerous isoforms that were differential when the gene was not. We have now added short vignettes to the manuscript discussing the biological relevance of these markers. Specifically we discuss the relevance of isoform marker *Oxr1-204* in marking Glutamatergic neurons, *Snap25-202* in marking L6b cells, and *Stxbp2-207* in marking a cluster of the L6b cells.

We have also added an analysis that searches for previously unannotated cell types that have isoform markers that were not identified with the gene clustering method. We show that the *App-205* isoform splits the L6 CT Grp_1 cluster with a much larger effect size than the gene. Indicating high isoform expression variability within the cell type and

providing supporting evidence that isoform resolution aids in refining cell types when gene quantifications alone cannot.

Lastly we have added a new analysis that looks at isoform expression between males and females in the SMART-Seq data. After searching in all subclasses for autosomal isoforms that best mark male versus female cells we found that the Vip subclass was marked by differential expression of *Shank1-203*.

A challenge in the use of scRNA-seq to assess isoform abundance is the limited capture efficiency of mRNAs, which results in the problem of gene dropouts (failing to capture any reads from a gene that is expressed), which is even more serious at the level of mRNA isoforms which are expressed at levels that may be a small or large fraction of the expression of the corresponding gene, and which require reads to cross the particular splice junctions that distinguish them from related isoforms. More explicit attention to issues of capture efficiency and coverage (see below) would make the results more convincing.

The SMART-Seq dataset generated 15,229,289,828 sequencing reads with an average of 2,419,268 reads per cell. Since every analysis involving SMART-Seq was aggregated at the cluster/subclass/class level, we looked at the number of reads contained within each cluster as it represents the fewest number of reads used in our analysis since all isoform markers are generated by comparing cells between classes, subclasses and clusters.

Comparing clusters against each other is equivalent to comparing multiple (pseudo) bulk RNA-seq samples to each other. We looked at the number of reads per cluster and found that the minimum number of reads per cluster was 9,499,100, the maximum was 2,264,376,507 and the median was 109,875,084. Since we are performing differential expression on the clusters, we can think of each cluster as a (pseudo) bulk RNA seq sample, and from this point of view the read depths in our “samples” are comparable or better than what is typical in RNA-seq. We also note that other papers report high coverage and capture efficiency for SMART-seq on real and simulated datasets (Westoby et al. 2018), and validate the accuracy of isoform quantifications. We quote a main result from Westoby et al. below:

[Redacted]

From Figure 2a from (Westoby et al. 2018)

“However, the extremely high recall of all the isoform quantification tools considered in this benchmark means that the overwhelming majority of isoforms from which reads are captured will be called as expressed.”

Additionally

"Our simulation-based analyses have demonstrated that Kallisto, Salmon, Sailfish, and RSEM can accurately detect and quantify isoforms in scRNA-seq to nearly the same accuracy as bulk RNA-seq data... Taken together, our findings show that isoform quantification is possible with scRNA-seq for SMARTer and SMART-seq2 data."

Our tools have been shown to be appropriate for quantifying isoforms. Isoform counts are assigned using kallisto's equivalence classes and the expectation maximization algorithm. The pseudoalignment step retains information about alignment across genomic locations and the expectation maximization algorithm assigns counts to transcripts in an iterative fashion such that the log likelihood of the count assignments are maximized (Bray et al. 2016).

In one simulated bulk RNA-seq dataset of 50 million reads based on the Human Brain Reference RNA HBRR-C4 dataset our tools demonstrated very low mean absolute relative differences and a high pearson correlation across all isoforms (Zhang et al. 2017). The figure is reproduced below.

[Redacted]

Figure 2 from (Zhang et al. 2017)

In summary, we believe that our datasets have sufficient read coverage when grouped together, as was done for every analysis step we have performed, and that our tools are accurate and properly assign counts to isoforms in a manner that minimizes dropout.

Cross-platform comparisons are also used to assess the strengths and weaknesses of different approaches, a worthwhile technical contribution. The manuscript is written clearly and the authors have done a commendable job in ensuring the reproducibility of their work by providing clear methods and links to code, etc. However, what is almost completely lacking from the manuscript is some sense of the biological importance of the descriptions of differential isoform usage across cell types and of the inferences the authors make – the paper feels descriptive and lacking in context. The main conclusions relate to which combinations of methodologies are best for scRNA-seq analyses, and which isoforms can be used as markers of particular cell types, but these conclusions are likely to be of interest primarily to specialists. No attempt is made to draw biological conclusions that might be of interest to a broader audience.

Thank you for the comments. Our paper originally lacked detailed biological conclusions and per your suggestion we have added them. As mentioned above, we have identified novel isoform markers that appear to be sex-specific. The *Shank1-203* isoform effectively distinguishes male and female cells within the Vip subclass a finding that refines previous data showing that *Shank1*, which has been shown to localize in Purkinje cells in the cortex (Böckers et al. 2004), is a sex specific gene whose expression is regulated by sex hormones (Berkel et al. 2018). Also, as mentioned above, we have developed a methodology that searches for isoform specific expression across the depth of the mouse primary motor cortex. While we have identified significant isoforms that vary across the depth, using a linear model, we believe that non-linear models may be better suited. We have also found numerous isoform specific markers for cell-types where the gene does not mark the cell-type. We have added these vignettes to the main manuscript.

Our findings also present, for the first time, a collection of isoform markers for cell types that are physically located within the mouse primary motor cortex. The scale and breadth of this transcriptomic “atlas” will certainly be useful to a general audience. To further make our findings available and accessible, we have created a searchable database for the isoform atlas (Supplementary Table 9). A user can search for cell types of interest and immediately know the set of isoforms that mark that cell type as well as the location of the cells for that cell type.

Specific issues:

1. Recent work has challenged the applicability of most scRNA-seq datasets to reliably estimate isoform abundance from all but the most highly expressed genes (<https://www.biorxiv.org/content/10.1101/2019.12.19.883256v1>), showing by simulation that presence of multiple isoforms in a single cell is often obscured by limited coverage resulting from insufficient mRNA capture efficiency in scRNA-seq library preps. Further, this study shows that isoform abundance can only reliably be estimated from scRNA-seq data by modeling coverage of each gene based on estimates of mRNA capture efficiency and expression level to assure sufficient depth of coverage. More attention should be given to these issues to convincingly show that the genes and datasets analyzed have sufficient coverage to reliably estimate isoform abundance.

We are aware of the preprint you linked to, and agree with its contents. A key point of the (Buen Abad Najar et al. 2019) paper is the claim that alternative splicing cannot be performed on individual cells. We acknowledge that studying individual cells may require modelling capture efficiency. However, when aggregating cells to perform differential expression, as we do in our manuscript, such a step is not necessary as the aggregate of a collection of cells is similar to deeply sequenced (pseudo) bulk RNA sequencing and as such contains sufficient information to perform differential expression at the isoform level. Nowhere in our manuscript do we make claims about isoform abundance *in individual cells*, but rather focus on clusters (or subclasses, and classes). In all of our analyses we have aggregated cells and this provides substantial additional power.

The depth of our pseudobulk is substantial: the SMART-Seq data was sequenced to an average depth of 2,419,268 reads per cell which is sufficient to accurately detect genes (Ziegenhain et al. 2017) and when aggregated at the cluster level as discussed above is sufficient to accurately quantify isoforms (Zhang et al. 2017; Westoby et al. 2018). The figure below (new Extended Data Fig. 15a) shows the distribution of reads *per cluster* in our dataset. This is an order of magnitude *higher* than typical bulk RNA-seq experiments.

Additionally we are using SMART-Seq2 developed by (Picelli et al. 2014) which improved on the original SMART-Seq protocol in numerous ways. One in particular is the uniformity of read coverage across transcripts for all genes (Supplementary Figure 9a in their paper). Furthermore, (Seirup et al. 2020) find that SMART-seq's higher sensitivity and read-depth allow for analysis of lower expressed genes and isoforms in their biological system, identifying isoforms with distinct expression in cases where 10x and MarsSeq could not.

2. In the section accompanying Fig 3 the authors use single-cell data to note that of the two isoforms of the Pvalb gene, only one (Pvalb-201) is expressed in the MOp region of the brain, and then extrapolate based on cell-type markers to map the isoform expression patterns onto spatial expression data. Given that only one of the two isoforms is expressed at a detectable level, the extrapolation reduces to the inference that any cell which expresses Pvalb expresses the Pvalb-201 isoform. That is, the authors are only mapping the spatial patterns of Pvalb isoform expression by virtue of the fact that it is the only isoform detectably expressed. This seems like a reasonable extrapolation, but it is not clear that it provides any clear advantage in terms of cell type classification over the use of the Pvalb gene as a marker, or other desirable outcome. Can the authors explain why this extrapolation adds something important, or provide a more compelling example of where an important biological inference is enabled by this sort of extrapolation?

Each method of transcriptome measurement yields different information: 3' capture allows for cell-type identification, full length allows for isoform identification, and spatial capture enables positioning the cell in space. By identifying *Pvalb-201* as the specific isoform

being expressed in the Pvalb subclass in the SMART-Seq data and then, by extrapolation, assigning the specific isoform expression to the Pvalb cells in the MERFISH data, we are linking isoform expression with spatial with a known spatial location. We argue that inferring spatially-resolved cell-type-specific isoform expression is critical for multiple reasons:

- Spatial RNA sampling is limited to a panel of genes (Wang et al. 2020), identifying isoform markers may aid in expanding spatial tag panels and in cell type identification and functional relevance in tissue
- Inferring cell-type specific isoform expression may reveal spatial patterns of alternative splicing, providing insight into the mechanism of cell signaling (Li et al. 2020) and spatial isoform regulation (McMillan et al. 2008) during brain development
- Spatial isoform markers enable more targeted assays for "automatic expression histology", for example new genetic tools are now available to knock-out isoforms (Thomas et al. 2020). A spatially resolved cell type specific atlas therefore provides actionable hypotheses, with inferred locations of expression that can help guide follow-up targeted assays.

We believe that spatially resolved isoform expression will also help in future image based screening techniques (He et al. 2020) where specific isoform markers are gleaned from paired spatial and isoform level RNA sequencing data that would otherwise be hidden at the gene level.

3. The figures are somewhat difficult to follow and need better legends. For example:

a. In Fig1e and e what do the dotted lines and circles represent? This should be explained in the legend.

Thank you for pointing this out. We have rewritten the figure legend to make it clearer that we are demonstrating, across the class, subclass, and cluster, differential isoforms for which the corresponding gene is not differential. We have also noted that the dotted lines correspond to the collection of cells that belong to the specific class, subclass, or cluster of interest.

b. Figure 2 and Ext Fig 8 are somewhat visually overwhelming. In interpreting the 289 distributions displayed as violin plots, what should the reader focus on? Most readers will focus on the centers of the distributions, or just the presence or absence of expression. Further, the variable y-axes and that the bulk of many distributions (and their means) are cut off by the axis limits makes this plot difficult to read. If the mean expression level is the most important piece of information, a heat map might be easier for readers to digest.

We agree that there is a lot of information contained within the violin plots and believe that it is important to show the expression of each isoform across all clusters. To make

the visualization more clear we have rewritten the legend to better guide the reader. As a supplement to these plots we have also made heat maps that highlight the normalized mean expression of each isoform relative to each cluster (new Extended Data Figure 10b,c).

c. The legend to Figure 3a appears to apply to 3c, while the legends to 3b and 3c appear to apply to 3a and 3b.

Thank you for pointing this out, we have fixed it.

d. In Figure 4c, the isoforms should be labeled for clarity.

Thank you for pointing this out, we have fixed it.

Reviewer Reports on the First Revision:

Referee #3 (Remarks to the Author):

Booeshaghi present an approach for quantifying gene expression isoforms from the recent BICCN datasets related to the mouse motor cortex. The authors apply these methods to SMART-Seq and 10xv3 datasets and identify novel splicing isoforms that, when applied in the context of clustering analyses, reveal additional diversity among cells and cell types beyond what can be identified through gene-level analyses. The level of correspondence between isoform identification between both technologies, which are quite different, is interesting and reassuring. Lastly, the authors present an approach for spatial positioning of isoform information onto MERFISH-based spatial transcriptomics data.

This work is technically rigorous and the results are presented clearly. I commend the authors for their extensive efforts to make their analyses and code reproducible and publicly accessible. Given that this manuscript has already been through one round of extensive review (which I did not contribute to), I will focus my comments on the perceived novelty of the work.

Regarding the perceived impact of this work, it is difficult to justify publication of this paper in Nature. There is little in the paper regarding the functional or biological consequences of the splice isoforms identified in the manuscript. While this issue was brought up by other reviewers in the previous round of review (Reviewer #2 specifically), this has not been adequately or sufficiently addressed in the current manuscript. Efforts to link the observed isoform diversity with other indices of biological utility, for example, whether such isoforms are conserved in orthologous cell types in other species have been not pursued. Critically, there has been no demonstration of the functional utility of any of the novel splice isoforms through direct perturbation methods.

A second point regarding the novelty of this work is that this paper is but one of many pursuing an identical question using similar datasets. For example, in a recent study by Feng et al, recently published in PNAS, (<https://www.pnas.org/content/118/10/e2013056118>), took a similar approach to isoform identification using datasets collected by the Allen Institute but with four times the number of SMART-seq characterized cells.

I understand the value of "focusing" the analysis here on mouse motor cortex to be consistent with

other work from the BICCN, but for the purposes of isoform identification and cell type mapping, given the relative paucity of sampled cells in the motor cortex relative to other brain regions, this is more a hindrance than a benefit.

Author Rebuttals to First Revision:

Response to reviewers 2

We appreciate the recent reviewer comments, and below provide a point-by-point response describing some of the work we have performed in response to the questions.

Referee #3:

Remarks to the Author:

Booeshaghi present an approach for quantifying gene expression isoforms from the recent BICCN datasets related to the mouse motor cortex. The authors apply these methods to SMART-Seq and 10xv3 datasets and identify novel splicing isoforms that, when applied in the context of clustering analyses, reveal additional diversity among cells and cell types beyond what can be identified through gene-level analyses. The level of correspondence between isoform identification between both technologies, which are quite different, is interesting and reassuring. Lastly, the authors present an approach for spatial positioning of isoform information onto MERFISH-based spatial transcriptomics data.

This work is technically rigorous and the results are presented clearly. I commend the authors for their extensive efforts to make their analyses and code reproducible and publicly accessible.

In light of questions we have received about the BICCN data, and to facilitate cross-comparisons with other datasets (see response to questions below), we have gone a step further and now taken the time to make our notebooks runnable on Google Colab.

Given that this manuscript has already been through one round of extensive review (which I did not contribute to), I will focus my comments on the perceived novelty of the work.

Regarding the perceived impact of this work, it is difficult to justify publication of this paper in Nature. There is little in the paper regarding the functional or biological consequences of the splice isoforms identified in the manuscript. While this issue was brought up by other reviewers in the previous round of review (Reviewer #2 specifically), this has not been adequately or sufficiently addressed in the current manuscript. Efforts to link the observed isoform diversity with other indices of biological utility, for example, whether such isoforms are conserved in orthologous cell types in other species have been not pursued. Critically, there has been no demonstration of the functional utility of any of the novel splice isoforms through direct perturbation methods.

In order to obtain a deeper understanding of the links between the isoform diversity and orthologous cell types, which we agree would add an important dimension to the paper, we investigated several datasets and performed several comparisons which we detail below. First, the question of cell type and splicing orthology *across species* is complex. While the BICCN consortium has generated full-length single-cell RNA-seq data from the

mouse primary motor cortex, in other species the full-length single-cell RNA-seq is single-nuclei RNA-seq data (from the marmoset and macaque). We explored whether it would be tractable to assess whether isoforms in the mouse MOp are conserved in orthologous cell types in macaque and marmoset, but concluded this would be extremely challenging / impossible with the data currently at hand. While there has been some methods development specific to cross-species single-cell comparison (<https://internal-journal.frontiersin.org/articles/10.3389/fcell.2019.00175/full>), there are still several major methodological gaps, and much research is still required for such analysis to be effective and reliable. Specific to our challenge, in addition to identifying and selecting orthologous cell-types in the marmoset and macaque, questions of isoform diversity require annotations that, while developed in the mouse, are lacking in marmoset and macaque. While the macaque and marmoset annotations are improving (<https://science.sciencemag.org/content/370/6523/eabc6617>, <https://bmcbgenomics.biomedcentral.com/articles/10.1186/s12864-020-6657-2>) they are incomplete and may make such isoform level comparisons challenging. Additionally, single-nuclei assays (macaque / marmoset) yield a greater fraction of reads from intronic regions, with high sequence homology across the transcriptome, making it challenging to assign UMIs to specific splice isoforms and compare isoform specificity to standard single-cell assays (mouse) (<https://www.nature.com/articles/s41587-020-0465-8>). Thus, while we agree the suggested investigation is interesting, it is beyond the scope of our current paper.

Nevertheless, motivated by the question of isoform diversity across orthologous cell types, we decided to compare and contrast our findings with those from recently published data from the mouse isocortex (<https://doi.org/10.1016/j.cell.2021.04.021>). This dataset provides an exceptional opportunity to ask whether the isoform diversity we discovered, specifically isoform markers present in the absence of differential gene expression, are conserved across brain regions. To examine this we solicited the help of a rotation student in the lab (Nicholas Markarian) who, in order to answer questions about isoform diversity in the mouse isocortex, re-processed and analyzed the data of <https://doi.org/10.1016/j.cell.2021.04.021>.

We have confirmed that many of our findings do, in fact, replicate across brain regions. For example, in a comparison of the Isocortex and Hippocampal formation structure comprising nine regions (Hippocampal region; Retrosplenial area; Subiculum, prosubiculum; Entorhinal area, lateral and medial part; Temporal association areas, perirhinal area, entorhinal area; Prelimbic area, Infralimbic area; Agranular insular area; Orbital area) the Myl6-203 isoform is differentially expressed in the GABAergic class while the gene is not. This is consistent with our results from the MOp for the GABAergic class.

Figure 1: Example of a gene with an isoform specific to the GABAergic class. The *Myl6* gene abundance distribution in $\log_{1p}(\text{Transcripts per million (TPM)})$ units across cells and *Myl6-206* isoform distribution in $\log_{1p}(\text{TPM})$ units across cells. The violin plots of the *Myl6* gene and *Myl6-206* isoform distributions show that the gene is not differential but the isoform is.

Moving down the hierarchy, we have similar results for the *Aldoa-203* isoform in the *Sst* subclass. The isoform is differentially expressed while the gene is not.

Figure 2: Example of a gene with an isoform specific to the Sst subclass. The *Aldoa* gene abundance distribution in log₁₀(Transcripts per million (TPM)) units across cells and *Aldoa-203* isoform distribution in log₁₀(TPM) units across cells. The violin plots of the *Aldoa* gene and *Aldoa-203* isoform distributions show that the gene is not differential but the isoform is.

These preliminary anecdotal results show conserved isoform splicing in matching cell types in different brain regions, though further analysis is required to fully understand these similarities, as well as possible differences. We've begun detailed analyses of these other brain regions comprising 24 notebooks, but as the analysis has ballooned in scope and complexity, we feel it is beyond the scope of this paper.

We agree that direct perturbation methods would be useful and informative to demonstrate the functional utility of novel splice isoforms. Validation studies with assays such as pgFARM (<https://www.nature.com/articles/s41588-019-0555-z>) are one of the BICCN's high priority research areas, and will undoubtedly lead to interesting future results.

A second point regarding the novelty of this work is that this paper is but one of many pursuing an identical question using similar datasets. For example, in a recent study by Feng et al, recently published in PNAS, (<https://www.pnas.org/content/118/10/e2013056118>), took a similar approach to isoform identification using datasets collected by the Allen Institute but with four times the number of SMART-seq characterized cells.

We are aware of this paper which came out while our paper was in review and we are excited that more researchers are focusing on isoform diversity and its implications for understanding cell types.

The Feng et al. paper studies alternative splicing events in the various cell-type hierarchies using full-length single-cell RNA-seq (SMART-Seq assay) from 17,222 cells from the primary visual cortex (VISp) and 10,068 cells from anterior lateral motor cortex (ALM). Their analysis is focused on isoform markers and alternative splicing events that can be derived from full-length RNA-seq data and they validate their results with bulk RNA-seq. Our analyses use full length SMART-Seq, 3' UMI 10xv3, and spatial MERFISH assays in numerous novel ways:

- 1. We use the full-length SMART-Seq data to find isoform markers for all collections of cells within the cell-type hierarchy. We find isoform markers that would be masked at a gene level analysis, and find multiple examples of transcriptional and post-transcriptional programs. This is an important advance that may help researchers characterize cell-types with more granularity.**
- 2. We show how isoforms could be used to refine cell-type annotations with a higher average effect size than standard gene expression. This addresses one of the major questions posed by the BICCN consortium: *What level of granularity of cell type definition is required for understanding the function of a given neural circuit?***

3. We perform extensive isoform and gene-level validation between all assays demonstrating that our quantifications are consistent. This is a novel advancement over Feng et al paper which simply validates against bulk RNA-seq.
4. We develop a framework and show, by way of example, how MERFISH spatial RNA-seq data can be used to infer spatial isoform expression for isoforms identified with the SMART-Seq data on the cell-types derived from the 10xv3 data. We believe this framework will be useful for developing, analyzing, and interpreting future single-cell spatial RNA-seq datasets.
5. As an example of how to use this framework, we show how isoform expression varies across the depth of the MOp.

We are excited about the future of isoform level biology and believe that our paper makes numerous novel advances over existing published literature and provides a coherent conceptual framework for studying the brain. It's likely that other brain regions exhibit similar patterns of isoform expression and analyzing these brain regions in a multi-assay approach will produce stronger and more detailed conclusions about splicing biology.

I understand the value of "focusing" the analysis here on mouse motor cortex to be consistent with other work from the BICCN, but for the purposes of isoform identification and cell type mapping, given the relative paucity of sampled cells in the motor cortex relative to other brain regions, this is more a hindrance than a benefit.

We agree that there are limits to this BICCN effort, which serves as a pilot and precursor to studying the whole mouse brain. The BICCN consortium initial focus on the mouse primary motor cortex limits not just our study but others as well, and yet it has been essential in establishing cross-consortium standards and protocols. As more data continues to be collected, the reproducibility standards we have enforced in our project, via fully functional shared notebooks that can reproduce results from raw data, the isoform atlas we have established will continue to evolve and be better understood in the context of other multimodal brain data that is being collected not just by BICCN, but around the world.